# A hybrid transistor with transcriptionally controlled computation and plasticity

Yang Gao[1], Yuchen Zhou [2,3], Xudong Ji [4,5], Austin J. Graham [1,6], Christopher M. Dundas[1,7], Ismar E. Miniel Mahfoud[1], Bailey M. Tibbett[1], Benjamin Tan[3,8], Gina Partipilo [1], Ananth Dodabalapur[2,3], Jonathan Rivnay [4,5] & Benjamin K. Keitz [1] ✉

Organic electrochemical transistors (OECTs) are ideal devices for translating biological signals into electrical readouts and have applications in bioelectronics, biosensing, and neuromorphic computing. Despite their potential, developing programmable and modular methods for living systems to interface with OECTs has proven challenging. Here we describe hybrid OECTs containing the model electroactive bacterium *Shewanella oneidensis* that enable the transduction of biological computations to electrical responses. Specifically, we fabricated planar p-type OECTs and demonstrated that channel de-doping is driven by extracellular electron transfer (EET) from *S. oneidensis*. Leveraging this mechanistic understanding and our ability to control EET flux via transcriptional regulation, we used plasmid-based Boolean logic gates to translate biological computation into current changes within the OECT. Finally, we demonstrated EET-driven changes to OECT synaptic plasticity. This work enables fundamental EET studies and OECT-based biosensing and biocomputing systems with genetically controllable and modular design elements.

Devices that transduce and amplify biological and chemical activity into electrical signals are highly desirable in a number of fields including sensing[1,2], neuromorphic computing[3], cellular computing[4], and wearable electronics[5]. For several of these applications, organic electrochemical transistors (OECTs) have emerged as ideal devices owing to their use of aqueous electrolytes, compatibility with biological systems, and low operating voltages[6,7]. In contrast to conventional electronics that rely on semiconducting and dielectric materials, OECTs utilize ions within an electrolyte to alter the doping state and conductivity of an organic mixed ionic-electronic conducting channel[8]. Because the entire volume of the channel is accessible to ions in the electrolyte, a relatively small potential change at the gate can significantly alter the channel's conductivity, giving OECTs exceptional

transconductance and sensitivity. In addition to sensing and flexible electronics applications, OECTs are promising devices for neuromorphic computing because synaptic weight, usually defined as channel conductance, can be altered by controlling ion transport in and out of the channel[9,10]. Overall, the inherent ability of OECTs to couple ionic and electronic transport makes them ideal devices for merging aspects of biological and traditional computation.

While ionic diffusion into the channel is typically controlled using an applied voltage at the gate electrode, biological or reduction-oxidation (redox) reactions in the electrolyte can also change the channel doping state. In effect, redox reactions can function as a secondary gate in the OECT. For example, lipid bilayer functionalization of the channel or gate followed by insertion of gated ion channels,

[1]McKetta Department of Chemical Engineering, University of Texas at Austin, Austin, TX 78712, USA. [2]Department of Electrical and Computer Engineering, University of Texas at Austin, Austin, TX 78712, USA. [3]Microelectronics Research Center, University of Texas at Austin, Austin, TX 78758, USA. [4]Department of Biomedical Engineering, Northwestern University, Evanston, IL 60208, USA. [5]Simpson Querrey Institute, Northwestern University, Chicago, IL 60611, USA. [6]Department of Pharmaceutical Chemistry, University of California San Francisco, San Francisco, CA 94158, USA. [7]Department of Biology, Stanford University, Stanford, CA 94305, USA. [8]Department of Chemistry, University of Texas at Austin, Austin, TX 78712, USA. ✉e-mail: keitz@utexas.edu

nanobodies, or other biomolecules allow OECTs to sense a variety of chemical and biological stimuli[11,12]. Similarly, redox-active enzymes, such as lactate or glucose oxidase, can directly transfer electrons into the channel to tune its conductivity in response to chemical substrates[13]. These applications also highlight the usefulness of OECTs for sensing and diagnostic applications, but achieving more complex sensing is challenging because individual enzymes, proteins, and other biomolecules are only capable of limited computation on a single device. In contrast, living cells perform a variety of extremely complex and robust computations that could potentially be tied to an OECT output. For example, bacteria can be engineered to perform computations including Boolean logic operations[14,15], analog/digital signal processing[16], cellular computing[4], and neuromorphic computing[17]. While OECTs have long been used to detect bacteria or sense the presence of specific metabolites[18], coupling more advanced computations to an electrical output via genetic circuits that regulate protein expression, small molecule synthesis, or other outputs that reliably interface with an OECT has proven challenging.

One promising strategy for interfacing bacterial computation with OECTs and other electronic devices is the use of electroactive bacteria. While all bacteria regulate ion flux into and out of the cell, electroactive bacteria can directly modulate electron transport across the cellular membrane in a process known as extracellular electron transfer (EET). Under anaerobic conditions, electroactive bacteria couple their central carbon metabolism to the oxidation or reduction of metal species in the environment via EET. Although naturally-occurring metals and metal oxides are the most well-studied electron acceptors for EET, synthetic materials including nanoparticles[19], conducting polymers[20], and a variety of electrode materials can also accept electron flux from these bacteria[21]. The use of electroactive bacteria in microbial fuel cells and similar bioelectronic devices has been extensively studied for power generation applications[22]. More recently, advances in synthetic biology and the engineering of model electroactive bacteria, such as *Shewanella oneidensis* and *Geobacter sulfurreducens*, have enabled broader applications of EET in sensing and biological computation[23,24]. To facilitate this transition, we, and others, have created genetic circuits that tightly regulate the expression of EET-relevant genes, allowing EET flux to be turned on and off in response to specific combinations of chemical, biological, and physical stimuli[25,26]. Thus, genetic regulation over electron transport via EET has the potential to serve as a universal interface between bacteria and electronic devices, including OECTs. Electrochemical transistors inoculated with electroactive bacteria can be conceptualized as dual-gate devices, wherein the initial gate represents the conventional electrode. The modulation induced by the second gate arises from electrochemical interactions facilitated by the bacterial cells. These interactions influence the charge balance within the channel, consequently altering its doping states.

Relative to conventional bioelectrochemical cells, OECTs present several advantages for measuring EET. Since OECTs can operate without the need for biofilms[27], which are essential for traditional bioelectrochemical systems, OECTs have quicker response time and more controlled and reproducible working environments. OECTs capitalize on small biological signals, which typically yield working electrode currents in the nanoampere range[28], to modulate output currents on the order of milliamperes, thereby producing significantly stronger signals[29]. This substantial signal enhancement enables detection with simple, economical electronic instruments. Consequently, OECTs are particularly well-suited for applications in point-of-care or resource-limited environments and hold promise for widespread commercial deployment due to their accessibility and cost-effectiveness[30]. With appropriate channel design and functionalization, OECTs can be made highly selective to specific chemicals, proteins, or metabolites, offering selective detection that might be harder to achieve in larger-scale electrochemical cells[31]. Finally, the ease of

fabrication and use also make OECTs suitable for high-throughput applications such as 96-well microplate screening[32]. Despite these advantages, instances of directly coupling electroactive bacteria with OECTs remain limited. Méhes et al. presented a notable example of real-time cellular EET activity monitoring with p-type OECT[18]. Using *S. oneidensis* (wild-type strain, MR-1) and the inherent amplification of OECT, changes in cellular metabolism beyond the limit of the conventional electrochemical setups were detected. While this work established an important proof of principle that EET could be detected using OECTs, we hypothesized that developing an improved mechanistic understanding of interactions between bacteria and the OECT combined with genetic regulation over EET flux could enhance our understanding of bacteria-OECT interactions, couple biological computation to an electrical response, and drive programmable changes to OECT synaptic plasticity.

Here, we developed hybrid transistors consisting of genetically engineered electroactive bacteria in planar p-type poly(3,4-ethylenedioxythiophene):poly(styrenesulfonate) (PEDOT:PSS) OECTs. Using *S. oneidensis* as a model EET-capable species, we first determined that channel conductance can be altered via the number and metabolic state of cells growing in the electrolyte. Next, we unraveled the biological and chemical mechanisms responsible for de-doping and current changes within the PEDOT:PSS channel using a combination of electrochemical and spectroscopic methods. To further illuminate the de-doping mechanism, we deployed genetically engineered *S. oneidensis* strains to regulate EET flux and analyzed the corresponding OECT outputs. Leveraging this mechanistic information, we converted EET flux from mutant strains carrying genetic Boolean logic circuits to electrical readouts, allowing the OECT to detect complex combinations of environmental signals. Finally, we characterized the synaptic behaviors of hybrid OECTs containing *S. oneidensis* strains, showcasing tunable synaptic weight tied to transcriptional outputs. Overall, our work contributes to the field of biosensing and biocomputing by augmenting OECT performance with genetically controllable inputs.

## Results

### Device characterization with *S. oneidensis*

Our planar OECTs were fabricated on quartz microscope slides with PEDOT:PSS-coated channel regions and Ti/Au gate, source, and drain electrodes (Fig. 1a, b, and Fig. S6). We first verified that *S. oneidensis* MR-1 could grow in the electrolyte (Shewanella Basal Medium, Table S2) and colonize the OECT under anaerobic conditions. OECT electrodes were constantly biased at $V_{GS} = 0.2\,V$ and $V_{DS} = -0.05$ and fluorescence microscopy images were taken 24 h post inoculation to determine cell viability and distribution within the device (Fig. 1c and 1d, Fig. S1a, b). Cells maintained a viability of $67 \pm 14\%$ and were found near the gate, channel, source, drain, and spaces in between (Fig. S1c, d). To corroborate these results, we monitored colony forming units (CFUs) and optical density at 600 nm ($OD_{600}$) within the OECT electrolyte. CFU counts were measured 24 h post inoculation in OECTs under constant $V_{DS} = -0.05\,V$ and $V_{GS}$ at $-0.5\,V$, $-0.2\,V$, $0.0\,V$, and $0.2\,V$. The $OD_{600}$ readings were measured in OECTs under constant $V_{DS} = -0.05\,V$ and $V_{GS} = 0.2\,V$. Consistent with our cell viability measurements, CFU counts dropped slightly after 24 h and showed no dependence on the gate potential (Fig. 1e). Similarly, $OD_{600}$ readings remained consistent during OECT operation, indicative of minimal cell growth over the 24-h period (Fig. S1f). When fumarate was included as a soluble electron acceptor to support anaerobic growth, cell viability improved to $80 \pm 10\%$ after 24 h post inoculation (Fig. S1c, e). As expected, the presence of fumarate also resulted in significant cell growth within the OECT, as indicated by increased CFU counts and $OD_{600}$ values (Fig. S1g, h). Taken together, these results indicate that PEDOT:PSS in the OECT can support cell maintenance, but not robust growth. While fumarate facilitated cell growth, we did not include it in

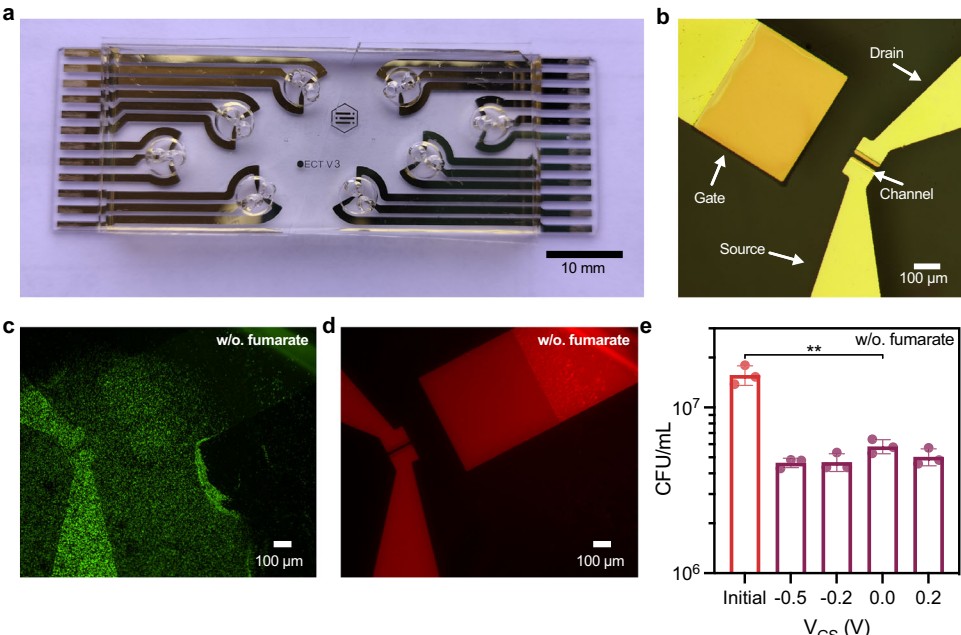

**Fig. 1 | Photo and microscopy images of the OECT. a** Eight OECTs on a microscope slide with PDMS sheets to form the OECT chambers, and **b** top view of a single OECT. **c**, **d** Representative fluorescence microscopy images of cells stained with LIVE/DEAD® BacLight™ cell assay with **c** live cells shown in green, and **d** dead cells shown in red. Background fluorescence signals from the gold electrodes (red channel) are also visible. Cells were supplemented with 20 mM lactate and no additional electron acceptor. **e** Colony forming units per mL (CFU/mL) were counted 24 h after OECTs operation with constant drain voltage $V_{DS} = -0.05$ V and gate voltages $V_{GS}$ biased at -0.5 V, −0.2 V, 0.0 V, or 0.2 V. CFU/mL $p$ value indicated = 0.0015. Cells were supplemented with 20 mM lactate and no additional electron acceptor. Data show the mean ± SD of 3 biological replicates, unpaired two-tailed Student's t-tests were performed without adjustments for multiple comparisons, n.s. represents $p > 0.05$.

subsequent experiments as its presence could discourage *S. oneidensis* from interacting with the PEDOT:PSS.

Under anaerobic conditions, *S. oneidensis* transfers electrons outside of the cell through the metal-reducing (Mtr) pathway, which is composed of three proteins MtrC, MtrA, and MtrB encoded on a single operon[33] (Fig. 2a). Based on previous work examining *S. oneidensis* on PEDOT:PSS-coated electrodes[20], we predicted that the bacteria could potentially influence channel current in the OECT through two major mechanisms (Fig. 2b). First, at positive gate voltages that energetically favor extracellular electron transfer, electrons may flow from bacteria cells to the gate. To maintain electrical neutrality, electrons will travel to the source via the external circuit. Subsequently, the electrons will flow out of or accumulate on the source electrode, where they can combine with PEDOT⁺ and de-dope the PEDOT:PSS channel. Alternatively, if the reduction potential of the channel is higher than that of the cell, the bacteria can directly reduce and de-dope the PEDOT:PSS, even in the absence of an applied potential at the gate. In either mechanism, biologically-driven de-doping of the PEDOT:PSS channel should be characterized by a pronounced decrease in channel current over time. Indeed, we found that devices inoculated with *S. oneidensis* MR-1 showed decreased channel current within 30 min relative to abiotic controls (Fig. 2c). Next, to examine device stability, OECTs were exposed to oxygen after 48 h of operation in the presence of *S. oneidensis*. The $I_{DS}$ recovered close to the original levels (Fig. S2a, b). To further examine material stability within the device, OECTs containing *S. oneidensis* were gently washed with soapy water and examined with atomic force microscopy (AFM). We found no significant changes in PEDOT:PSS film thickness compared to pre-inoculation films (Fig. S7a). Similarly, the channel surface roughness exhibited minimal alteration, with root mean square (RMS) roughness values of 2.4 nm and 2.6 nm for pre-inoculation and post-inoculation films, respectively (Fig. S7c, d). The PEDOT grain sizes were extracted from phase images (Fig. S7e, f), and subtle segregation of the PEDOT cores (brighter color) was

discernible from the post-inoculation samples with PSS filling the space in between (darker color)[34]. Importantly, the histograms for PEDOT grain size exhibited comparable distributions, suggesting minimal changes in the material constitution following bacterial incubation (Fig. S7b). Collectively, these results indicate good device stability and a reversible mechanism between *S. oneidensis* and OECT channel de-doping. Finally, to facilitate comparisons between different biological conditions and account for dynamic changes to channel current, the rate of current decay was fitted to a single exponential decay model to extract a rate constant characteristic of EET-driven channel current decreases (Fig. 2c).

On a per-cell basis, currents from EET flux are relatively small[35]. However, more cells should generate more flux and faster response times. As expected, the measured rate constant associated with the decrease in OECT current was proportional to the size of the starting cell population (Fig. 2d, Fig. S2c). Because EET flux is connected to the bacteria cells' central carbon metabolism, different carbon sources generate varying amounts of EET flux. Lactate is the preferred carbon source for *S. oneidensis*, followed by pyruvate, while acetate cannot be metabolized under anaerobic conditions[36]. As predicted, cells metabolizing lactate or pyruvate yielded the fastest current response while starved cells or those metabolizing acetate exhibited rates closer to abiotic controls (Fig. 2e, Fig. S2d). These results are strong indicators that cells remain metabolically active within the OECT and that cellular metabolic flux is correlated with channel de-doping. Furthermore, to confirm the channel current reduction is driven by live *S. oneidensis* MR-1 cells, OECTs were inoculated with heat-killed and lysed *S. oneidensis* MR-1 cells, cell metabolic products from the supernatant of overnight *S. oneidensis* cultures, and live *E. coli* MG1655 cells that are incapable of EET. As depicted in Fig. S2e, f, only metabolically active and lysed *S. oneidensis* MR-1 caused significant $I_{DS}$ decay. Relative to living cell samples, the lysed samples exhibited a marked linear and slower $I_{DS}$ decay profile that is most likely a result of the release of

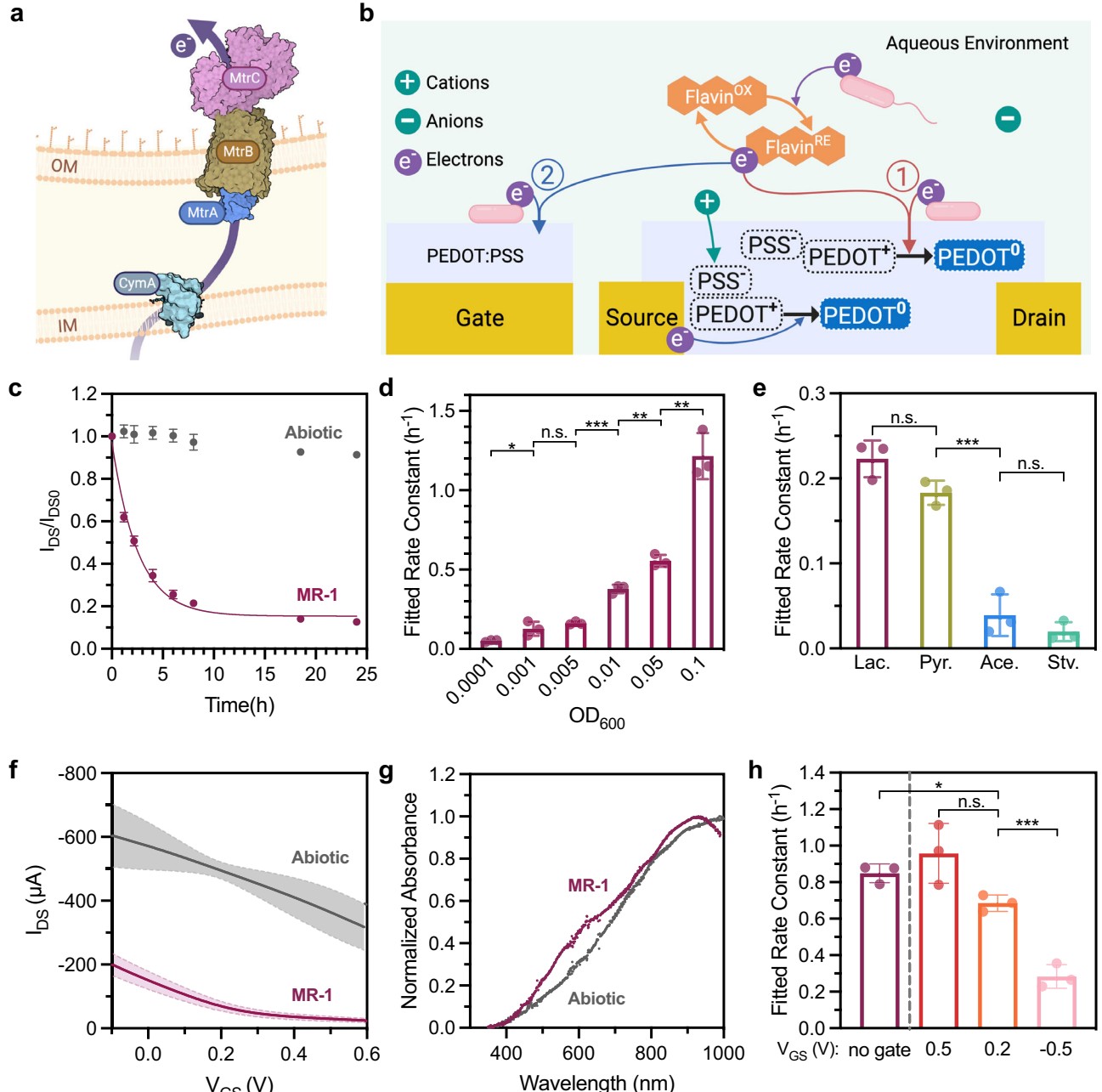

**Fig. 2 | The proposed cellular de-doping mechanisms and OECT output changes induced by *S. oneidensis* MR-1. a** Illustration of the MtrCAB pathway directing electrons to extracellular electron acceptors through the inner membrane (IM) and outer membrane (OM). **b** Proposed channel de-doping mechanisms through direct and mediated extracellular electron transfer (EET). **Path 1**, cellular metabolic electrons are transferred directly to the PEDOT:PSS channel. **Path 2**, electrons are transferred to the channel via the gate and external circuits (circuits not shown). **c** The $I_{DS}/I_{DS0}$ curve for OECTs inoculated with *S. oneidensis* MR-1, cells were supplemented with 20 mM sodium lactate and no additional electron acceptors. Initial inoculum $OD_{600}$ = 0.01. Fitted rate constants for *S. oneidensis* inoculated OECT with different **d** inoculation densities ($OD_{600}$) or **e** carbon sources. Samples with different inoculation densities were supplemented with 20 mM sodium lactate as the carbon source. Fitted rate constant $p$ values for pairs indicated from left to right $p$ = 0.0431,

$p$ = 2 × 10⁻⁴, $p$ = 0.0026, and $p$ = 0.0016. Carbon source and starvation samples were prepared at an inoculation $OD_{600}$ of 0.01 and supplemented with either 20 mM sodium lactate (Lac.), 20 mM sodium pyruvate (Pyr.), or 20 mM sodium acetate (Ace.). No carbon source was added to the starved cells (Stv.). Fitted rate constant $p$ value for the indicated pair $p$ = 9 × 10⁻⁴. **f** Transfer curves were measured in the presence and absence of *S. oneidensis*. The shaded region indicates the range of standard deviation. **g** UV–Vis spectra of the PEDOT:PSS channel in the presence and absence of *S. oneidensis*. **h** $I_{DS}$ decay profiles compared for OECTs biased at different gate voltages and gate-removed 2-electrode devices. Initial inoculum $OD_{600}$ = 0.01. Fitted rate constant $p$ values for pairs indicated from left to right $p$ = 0.0142, and $p$ = 9 × 10⁻⁴. Data show the mean ± SD of 3 biological replicates, unpaired two-tailed Student's t-tests were performed without adjustments for multiple comparisons, n.s. represents $p$ > 0.05. **a** and **b** created with BioRender.com.

redox-active components, such as thiols, NADH (nicotinamide adenine dinucleotide), quinones, cytochromes, and other reductants following cell lysis. Overall, these results demonstrate that de-doping of PEDOT:PSS and associated decreases in channel current are closely tied to cellular metabolism.

## Mechanism of OECT channel de-doping

Depletion mode OECTs are characterized by a decrease in channel current as the bias voltage at the gate becomes increasingly positive. Accordingly, we measured the transfer curves of devices inoculated with bacteria and observed a noticeable decrease in the channel

current and source electrode potential after 24 h of cell incubation in the device (Fig. 2f, Fig. S3a). While the applied gate and drain voltages were noted as $V_{GS}$ and $V_{DS}$ with respect to the source electrode, the effective gate voltages ($V_G^{eff}$) were determined by comparing the measured source voltages ($V_S$) against an Ag/AgCl pellet pseudo-reference electrode (RE), given by $V_G^{eff} = -V_S$. The measured decrease of the source electrode potential is indicative of reductions to the source and channel, which is in accordance with our hypothesis that *S. oneidensis* MR-1 can de-dope the channel. To compare the channel current decrease and electrode potential drop, we also measured the source and gate potential against an Ag/AgCl RE with fixed $V_{GS} = 0.2$ V and $V_{DS} = -0.05$ V (Fig, S3b). The strong correlation between decreasing $V_G$ and $|I_{DS}|$ (absolute) values also explains the changing $I_{DS}$ rate as cells retain metabolic activity and continue to perform EET within the OECT.

To further verify that the observed channel current reduction in the presence of *S. oneidensis* MR-1 was due to de-doping of the PEDOT:PSS channel, we conducted in-situ UV–Vis measurement using a modified OECT with a larger channel area (Fig. S3c). Previous studies have shown that the doping state of PEDOT:PSS can be characterized via UV–Vis spectroscopy[37]. Specifically, the neutral and polaronic states correspond to absorption peaks around 650 nm and from 800 nm to far infrared, respectively[38,39]. We initially measured spectra in abiotic devices using a non-polarizable Ag/AgCl pellet gate biased at varying voltages. As depicted in Fig. S3d, the neutral PEDOT:PSS peak at 650 nm increased as the gate voltage became more positive and the channel was increasingly de-doped. Correspondingly, the absorbance around 900 nm for the PEDOT:PSS polaron decreased sharply as the gate bias voltage increased from 0.6 V to 1.0 V. Next, large channel OECTs containing an Au gate were inoculated with *S. oneidensis* MR-1 (inoculation OD$_{600}$ = 0.01) and operated continuously with $V_{GS} = 0.2$ V and $V_{DS} = -0.05$ V. To account for absorption from cells, a channel-less OECT with the same inoculum and operation conditions was used as the blank before each spectrum measurement. As shown in Fig. 2g and Fig. S3e, the channel UV–Vis spectrum during the low $I_{DS}$ plateau (ca. 20.5 h after inoculation) displayed a similar profile to the de-doped abiotic channel with an Ag/AgCl gate biased at 0.5 V, indicating the presence of *S. oneidensis* cells had a comparable de-doping effect as an Ag/AgCl gate poised at this potential.

Because these EET de-doping experiments were conducted with cells present on both the gate and channel, the direct biological reduction of the channel could not be isolated from the observed decreases in current in devices containing *S. oneidensis* MR-1. Therefore, we also examined a two-electrode version of the small OECT where the Au gate electrode was completely removed and EET can only affect the channel doping state via direct reduction (Fig. S3f). Similar to our previous continuous OECT operation conditions, a constant $V_{DS} = -0.05$ V was applied to the 2-electrode devices. As shown in Fig. 2h and Fig. S3g, the 2-electrode devices exhibited pronounced $I_{DS}$ decay upon inoculation with *S. oneidensis* and a decay rate comparable to our three-terminal OECTs with $V_{GS} = 0.2$ V. The pronounced $I_{DS}$ change in 2-electrode devices confirmed the direct and potent interaction between bacteria cells and the PEDOT:PSS channel.

De-doping of PEDOT:PSS could also occur through interaction of *S. oneidensis* with the gate and subsequent electron accumulation on the source. To further examine this possibility, we compared the channel current-decay rates from our original three-terminal OECTs with different gate bias voltages. Gate voltages of 0.5 V, 0.2 V, and −0.5 V were selected, with decreasing energetic favorability for EET to the gate. As shown in Fig. 2h, the $I_{DS}$ decay rate constants increased with more positive gate bias voltages for biotic OECTs, while the abiotic OECTs showed negligible current change (Fig. S3g). The ability of gate bias potential to modulate the $I_{DS}$ rate constant confirmed our hypothesis that electrons transferred to the gate could participate in channel reduction via the external circuit. Together, these results

suggest that *S. oneidensis* MR-1 can directly de-dope the PEDOT:PSS channel via electron transfer, and the gate bias voltage plays a significant role in modulating the de-doping process. Although we were unable to completely distinguish between electron transfer at the gate and channel, the ability to tune the biological response based on the applied gate voltage highlights how multiple parameters, both biological and device-based, can control OECT performance.

## EET drives changes in OECT current output

Encouraged by our results showing *S. oneidensis* MR-1 could effectively de-dope the PEDOT:PSS channel, we investigated whether these changes could be attributed to EET. In *S. oneidensis*, EET occurs through direct or indirect means. Direct EET is mediated by cell attachment and interaction with a suitable redox-active surface by the Mtr pathway, while indirect EET is controlled by the biosynthesis and secretion of flavins, which act as soluble redox shuttles[40]. We employed several genomic deletion strains and protein expression tools to control EET flux and determine which of the above mechanisms might be contributing to the OECT response (Fig. 3a). Specifically, the △*bfe* strain has decreased extracellular flavin secretion due to the deletion of bacterial flavin exporter gene *bfe*, which is critical for indirect EET[41]. The △*lysis* strain has impaired biofilm formation due to the deletion of lysis operons *SO2966* to *SO2974*[42]. Finally, both △*mtrC* and △Mtr strains lack key EET proteins and exhibit decreased EET rates[43]. While the △*mtrC* strain has the key outer membrane EET proteins MtrC, OmcA, and MtrF deleted, the △Mtr strain has additional genomic deletions of periplasmic electron carriers, namely *mtrA*, *mtrD*, *dmsE*, *SO4360*, and *cctA*[43].

To determine the effects of the two EET pathways on OECT response, we first measured rate constants for $I_{DS}$ decay from devices inoculated with △*bfe*, △*lysis*, and wild-type *S. oneidensis* strains MR-1 (Fig. 3b). We observed no statistically significant differences between these three strains, suggesting biofilm formation and flavin secretion are not required for PEDOT:PSS de-doping. It is likely that on the time scale of our experiments (ca. 24 h), flavin secretion and extracellular concentration are not significant enough to affect device performance. However, adding exogenous flavins (flavin mononucleotide, 1 μM) did increase the rate of current decay for all strains tested (Fig. 3b), indicating under certain conditions indirect EET can be an important contributor to OECT performance. We also observed a dose-dependent response to increasing flavin concentration. Specifically, we measured a sigmoidal response of the channel current-decay rate constant with increasing exogenous FMN concentration (Fig. 3c), consistent with a direct EET transfer mechanism via FMN-bound MtrC[44]. The sigmoidal relationship between flavin concentration and current decay has previously been observed in some microbial fuel cells and is likely due to flavin saturation, which is exacerbated by the small volume and low surface area of the OECT electrodes in our devices[45].

Next, we measured current responses from the EET-deficient △*mtrC* and △Mtr strains. As expected, the rate constant and current-decay rate of these strains were significantly lower than those of *S. oneidensis* MR-1, indicating that impaired EET pathways significantly slowed the channel de-doping process (Fig. 3d, Fig. S8c). To further confirm that differences in channel de-doping were due to EET, we constructed plasmid vectors encoding key EET genes to complement EET-deficient mutants. Specifically, we constructed an *mtrC* Buffer gate by placing the *mtrC* gene downstream of the P$_{tacsymO}$ promoter and transformed it into the △*mtrC* strain (Fig. S8d, insert). For this circuit, the presence of IPTG [IPTG = Isopropyl β-D-1-thiogalactopyranoside] allows RNA polymerase to transcribe the *mtrC* gene and ultimately increase EET flux. We also constructed a *mtrCAB* Buffer gate regulated by OC6 [OC6 = 3-oxohexanoyl-homoserine lactone], which was transformed into the △Mtr mutant (noted as △Mtr+*mtrCAB*). Mutants harboring empty

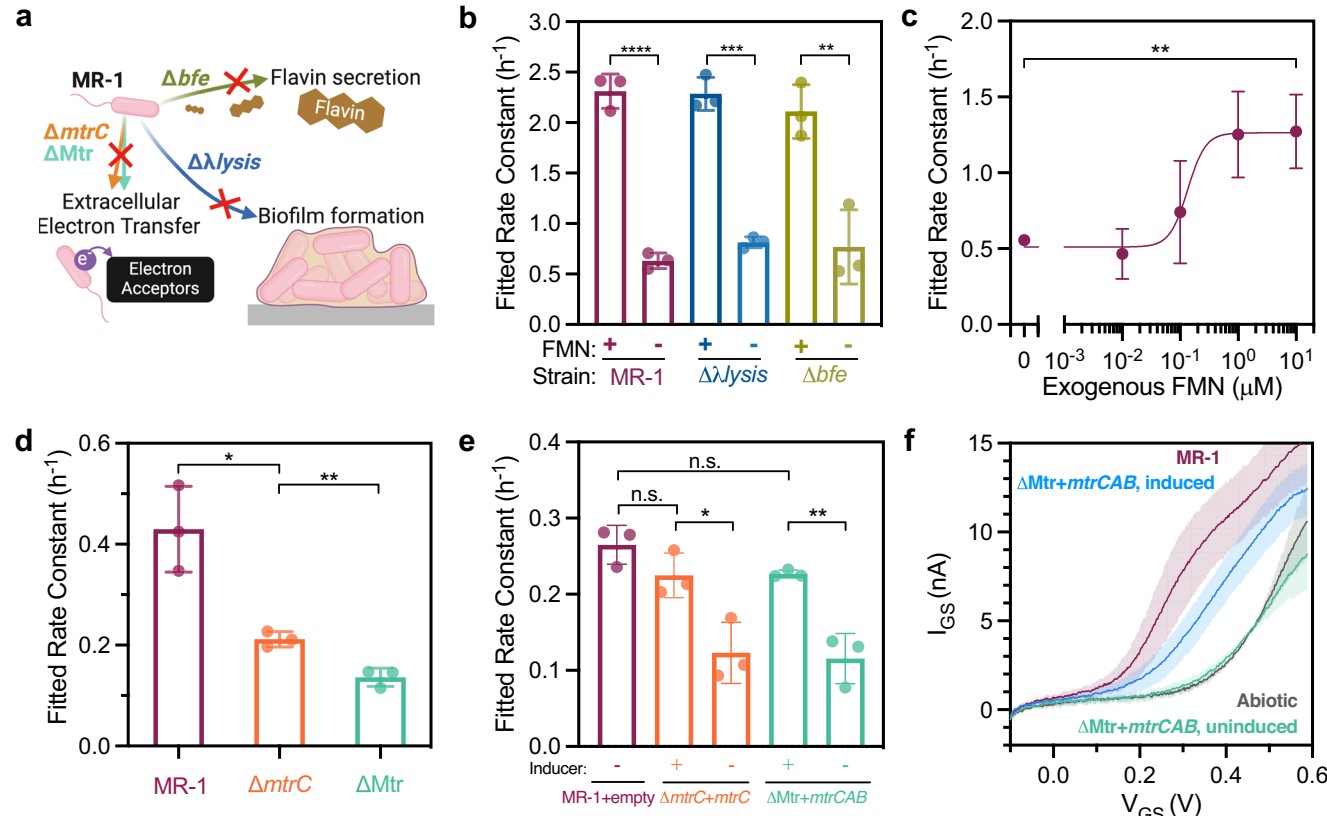

**Fig. 3 | OECT response to different EET mechanisms. a** Illustrations of the knockout strains with reduced secretion of flavin (△*bfe*) or impaired biofilm formation (△λ*lysis*). **b** Channel current $I_{DS}$ decay rate constants of knockout strains with and without 1 μM exogenous flavin mononucleotide (FMN). Fitted rate constant *p* values for pairs indicated from left to right $p = 9.9 \times 10^{-5}$, $p = 1 \times 10^{-4}$, and $p = 0.0068$. **c** $I_{DS}$ decay rate constants of *S. oneidensis* MR-1 supplemented with varying concentrations of exogenous FMN. Fitted rate constant *p* value for the indicated pair $p = 0.0073$. Inocula in (**b**) and (**c**) were adjusted to $OD_{600}$ of 0.05. **d, e** $I_{DS}$ decay rate constants for △*mtrC* and △Mtr knockout strains **d** without vector controls (noted as +empty) without the respective EET genes were used as negative controls. Prior to inoculation into OECTs, all strains were anaerobically induced for 18–24 h with 1 mM IPTG or 100 nM OC6 to ensure steady-state protein expression. As illustrated in Fig. 3e, mutants with restored EET gene expression yielded significantly higher current-decay rate constants relative to empty vector controls. The lower rate constants observed in the induced samples compared to wild-type *S. oneidensis* MR-1 are likely due to the absence of other outer-membrane EET proteins, such as OmcA and MtrF, in these genetic constructs. To further verify the electrochemical activity of the induced Buffer gates samples, gate currents were measured with gate voltage scans. As shown in Fig. 3f, △Mtr strains with induced *mtrCAB* exhibited higher oxidation currents compared to the uninduced control, which was almost identical to the abiotic baseline. Together, these data indicate that de-doping of PEDOT:PSS and the resulting current decreases are attributable to EET and specific EET-relevant protein expression.

vector plasmids and **e** complemented with *mtrC* and *mtrCAB* Buffer gates. Initial inoculum $OD_{600} = 0.1$ for **d**. Fitted rate constant *p* values for pairs indicated from left to right for **d** $p = 0.0118$ and $p = 0.0051$, **e** $p = 0.0238$ and $p = 0.0045$. **f** Gate current $I_{GS}$ from △Mtr knockout strains with induced and uninduced *mtrCAB* Buffer gates, measured at 24 h after inoculation. The shaded region indicates the range of standard deviation. Data show the mean ± SD of 3 biological replicates, unpaired two-tailed Student's t-tests were performed without adjustments for multiple comparisons, n.s. represents $p > 0.05$. **a** created with BioRender.com.

## Interfacing genetic logic gates with OECTs

Transistors, including OECTs, require circuit connections to perform more complex logic. In contrast, bacteria have been programmed to exhibit logic-based responses[46], analog computing[4,16], memory, and neuromorphic behavior[17]. For these applications, the desired computation is genetically encoded within each bacterium and the calculation is performed by the community. To determine whether similar genetic circuits could be connected to OECT performance and enable more complex logic on a single device, we evaluated transcriptionally controlled Boolean logic gates that regulate EET gene expression and associated flux in response to combinations of small molecule inputs. Specifically, we leveraged previously developed NAND and NOR transcriptional logic gates that control EET flux in response to combinations of inducer stimuli[25]. As building blocks in digital circuit design, the NAND and NOR gates could potentially enable more sophisticated genetic circuits. The two-input NAND and NOR gates regulate *mtrC* gene expression using tailored sensing blocks that respond to common small molecule inputs - IPTG and OC6 for the NAND gate, or OC6 and aTc [aTc = anhydrotetracycline] for the NOR gate (Fig. S4a, b). Plasmids encoding the NAND (pNAND-*mtrC*) and NOR gates (pNOR-*mtrC*) gates were transformed into △*mtrC* mutants. The strains were induced overnight to achieve steady-state gene expression levels. Following overnight induction, mutant strains were inoculated into OECT devices for continuous channel current monitoring (Fig. S9a, b). Subsequently, the corresponding decay rate constants were evaluated and plotted as 2D heat maps to present the response to varying inducer concentrations (Fig. 4a, b). As expected, the NAND and NOR gates both yielded current-decay rate constants conforming to the expected truth tables, demonstrating the direct conversion of transcriptional logic to electrical signals by the OECT. Additionally, we noticed similar decay profiles of the channel current $I_{DS}$ in mutants expressing logic '1' in the NAND and NOR gates. Conversely, the profile

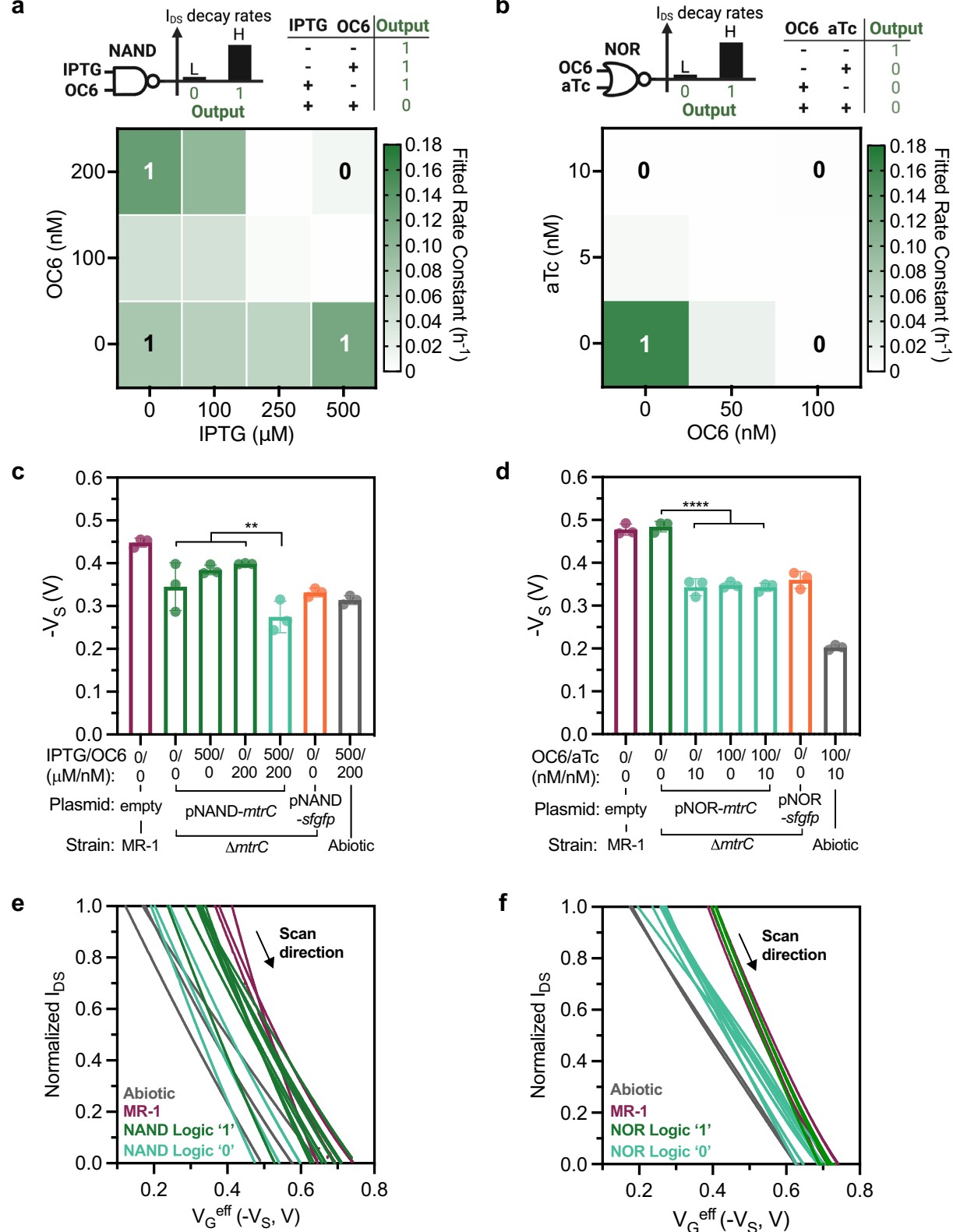

**Fig. 4 | The OECT responds to strains carrying genetic Boolean logic gates.** Channel current $I_{DS}$ decay rate constants plotted with combinatorial inducer (IPTG, OC6, and aTc) concentrations of △*mtrC* mutants carrying **a** NAND and **b** NOR Boolean logic gates. Measured source potentials $V_S$ were extracted at gate voltage $V_{GS}$ = 0 V for the **c** NAND and **d** NOR gate samples. Contrast tests for the $V_S$ showed *p* values for **c** *p* = 0.00204 and **d** *p* = 2.36 × 10⁻⁷. Channel current $I_{DS}$ were normalized to the range in the transfer curves for **e** NAND and **f** NOR gate samples. Data show the mean ± SD of 3 biological replicates, statistical analysis was conducted using general linear hypothesis tests to evaluate interaction terms and linear contrasts for logic 1 s and logic 0 s of the NAND and NOR logic gates. **a** and **b** created with BioRender.com.

for the logic '0' gate samples bore resemblance to that of negative controls lacking *mtrC* expression (Fig. S4e, f). Together, these results demonstrate that transcriptional logic can be coupled to an OECT output.

Similar to the de-doping mechanism investigation, we used an Ag/AgCl pellet RE to monitor the electrode potential with the negative of the measured source potential used as the effective gate potential, $V_G^{eff} = -V_S$. To ensure complete turn-on and turn-off behavior, mutants were induced using a combination of maximum and minimum concentrations of inducers, corresponding to the corners of the respective 2D heat map (Fig. 4a, b). For instance, 100 nM of OC6 and 10 nM of aTc were used for the NOR gate. The source potentials were measured 24 h after inoculation against Ag/AgCl RE with the applied $V_{GS} = 0$ V for all induced mutants and controls. As depicted in Figs. 4c, d, the mean source potential decreased by 101.4 mV and 139.6 mV for logic 1 s compared to logic 0 s for NAND and NOR gates, respectively. The shifts in source potential are clear evidence for reduction of the source electrode and channel by mutants successfully controlling EET flux according to the predicted circuit logic. Channel reduction in response to *mtrC* expression was further supported by transfer curves. To mitigate variations in transfer curve slopes resulting from channel thickness or fabrication inconsistencies (Fig. S4c, d), $I_{DS}$ values were normalized to the range of 1. As shown in Fig. 4e, f, we measured a clear shift toward more negative $V_S$ (or positive $V_G^{eff}$) for mutants expressing logic 1 s (NAND or NOR gates were on) and positive *S. oneidensis* MR-1 controls. Leaky gene expression and minimal EET activities from 'off' mutants could be the cause of the slight overall shift in the respective $V_S$ and transfer curves. Overall, the integration of transcriptional logic with OECTs creates a general and streamlined platform that unlocks the diverse computational power available to biological systems.

## Synaptic behavior in OECTs containing electroactive bacteria

In addition to sensing applications, OECTs have emerged as promising platforms for emulating synaptic plasticity due to their analogous working principle as synapses. In a biological synapse, the presynaptic action potential releases neurotransmitters across the synaptic cleft that modulate the postsynaptic membrane potential[47]. When operating as a neuromorphic element, an OECT uses the gate voltage as the presynaptic input and the channel current as the postsynaptic output (Fig. 5a). Because the channel current is a function of ion diffusion as well as redox processes within the electrolyte, the history of presynaptic inputs can influence channel conductance, resulting in a memory effect within the device. Thus, we explored whether *S.oneidensis* MR-1 could alter synaptic weight in an OECT in a genetically controlled manner.

A crucial property of synaptic transistors is their non-volatility, enabling them to effectively capture both short-term and long-term plasticity and mimic the dynamic and persistent nature of synaptic changes in biological neural networks. The short-term plasticity can be visualized as hysteresis in the transfer curve. At a channel voltage of $V_{DS} = -0.05$ V, we measured transfer curves of OECTs containing *S. oneidensis* cells by cycling the gate voltage between −0.5 V and 0.5 V. As shown in Fig. 5b, *S. oneidensis* MR-1 inoculated OECTs showed a marked hysteresis profile relative to abiotic controls, suggesting a form of memory endowed by the bacteria. Encouraged by this result, we evaluated the synaptic behavior of these devices. Two forms of short-term plasticity (STP), namely, paired pulse facilitation (PPF) and paired pulse depression (PPD) are commonly examined for their importance in decoding temporal information. As shown in Fig. 5c, the paired pulses were defined by the pulse amplitude $V_P$, pulse duration $t_P$, and pulse interval $\triangle t$. The PPF and PPD indices were defined as $A_2/A_1 * 100\%$, where the $A_1$ and $A_2$ represent the channel current pulse amplitude relative to the pre-pulse level, induced by the paired gate voltage pulses. For simplicity, the $A_2/A_1$ index will be used instead of PPF and PPD in the following sections. The synaptic weight was defined

as channel conductance (G) and the weight change was measured 30 s after the end of the last gate voltage pulse.

To compare the effects of pulsed gate voltages ($V_P$) on the channel responses and associated short-term memory behavior, the pulse duration $t_P$ and pulse interval $\triangle t$ were initially kept constant at 80 ms. As shown in Fig. 5d, abiotic OECTs displayed small and symmetrical weight changes to the varying gate pulse voltages. On the other hand, *S. oneidensis* MR-1 inoculated OECTs exhibited minimal conductance changes to negative gate pulses, while the conductance increased sharply with more positive $V_P$. The same asymmetric response to positive versus negative pulses was also observed in the $A_2/A_1$ index, where only a marked decrease was observed with more positive $V_P$ from biotic OECTs (Fig. S5a). Next, we tested the spike timing dependence by varying the gate pulse interval ($\triangle t$) while maintaining a constant pulse duration $t_P$ of 80 ms and pulse voltage $V_P$ of 0.5 V or −0.5 V. Interestingly, for *S. oneidensis* inoculated OECTs, the channel conductance increased with positive pulses across a range of spike timing (Fig. 5e), while the conductance changes were negligible with negative pulses (Fig. S5b). Consistent with our previous results, minimal and symmetrical conductance changes were observed in abiotic OECTs regardless of the pulse interval $\triangle t$ and pulse voltage. The $A_2/A_1$ index was also measured with different spike timing. For positive pulses, the $A_2/A_1$ index for all samples decreased exponentially as $\triangle t$ decreased, with biotic OECTs exhibiting more pronounced changes (Fig. S5c). Conversely, when responding to negative pulses the $A_2/A_1$ index for biotic OECTs was only distinguishable from the abiotic ones when the $\triangle t$ decreased to 80 ms (Fig. S5d). Together, the *S. oneidensis* MR-1 inoculated OECTs exhibited distinct synaptic behaviors, responding exclusively to positive gate voltages with a short-term increase in synaptic weight. Conversely, negative pulses induced negligible synaptic response in the biotic OECTs, comparable to the responses observed in abiotic OECTs.

To examine the reversibility of the synaptic behaviors and long-term plasticity, we subjected OECTs to continuous presynaptic stimuli. As depicted in Fig. 5f, the gate voltage was repeatedly pulsed with $V_P$ alternating between 0.5 V and −0.5 V for a total of 8 sessions. In each session, a pulse set of 4 was repeated 30 times for every 60 s, with the pulse duration $t_P$ and interval $\triangle t$ maintained at 55 ms. Conductance baselines were obtained by removing the spiking during the gate pulses for better visualization. The corresponding channel conductance for the *S. oneidensis* MR-1 inoculated OECTs exhibited consistent responses to repeated stimuli sessions (Fig. 5f, Fig. S5f). The conductance baselines for each continuous stimuli session were fitted with a one-phase exponential mode, while the positive session marked as S1, S3, S5, and S7 (Fig. S5g) and negative ones marked as S2, S4, S6, and S8 (Fig. S5h). The average time constants for positive and negative stimuli were 429.5 s ± 18.6 s and 1083.1 s ± 71.0 s, respectively (Fig. S5i). The small variation in the fitted time constants indicate consistent and reversible synaptic modulation of the *S. oneidensis* inoculated OECTs. In accordance with the paired pulse response, the saw-tooth channel current curves in Fig. S5g demonstrate the strong transient doping of the channel by the positive pulses and rapid de-doping during resting intervals ($V_{GS} = 0$ V). The rapid de-doping during resting also suggests minimal long-term plasticity (LTP) induced by the presynaptic pulse in the biotic OECTs. However, further experiments on cell metabolic rates and viability are necessary to understand the long-term stability and plasticity of the biotic OECT.

Finally, after establishing the basic synaptic behaviors of OECTs containing *S. oneidensis*, we examined the correlation between EET and synaptic modulation using EET-deficient knockout strains complemented with *mtrC* or *mtrCAB*. For these experiments, paired inputs of $V_P = 0.5$ V and $t_P = \triangle t = 80$ ms were used. EET-deficient mutant strains ($\triangle mtrC$ and $\triangle$Mtr) containing the appropriate plasmids were inoculated into OECTs following steady-state gene expression. As expected, strains with reconstituted Mtr pathways ($\triangle mtrC + mtrC$ and

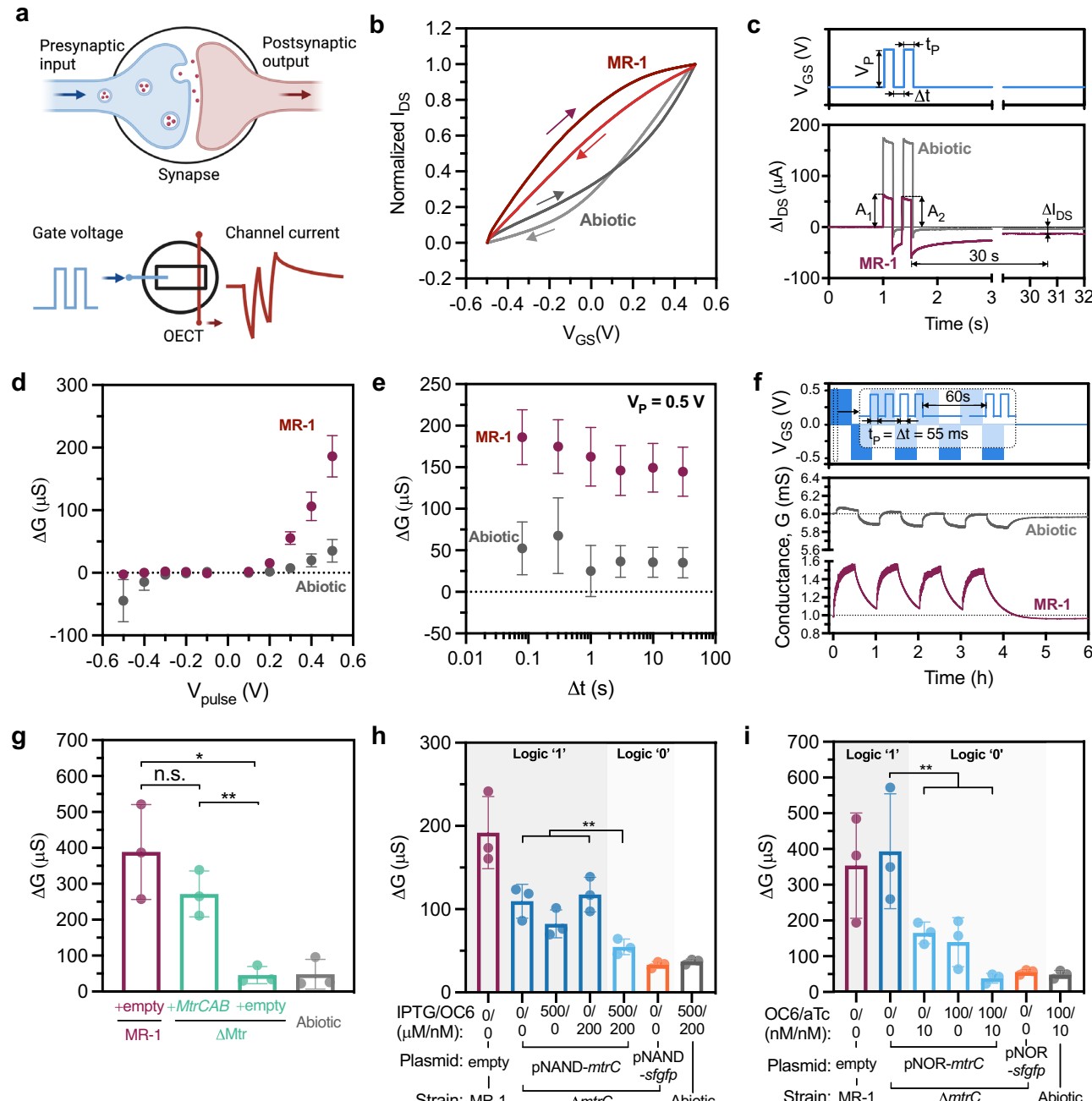

**Fig. 5 | Synaptic behaviors of OECTs inoculated with electroactive bacteria.**
**a** Analogous to the neuronal synapse, the OECT employs gate voltage as the presynaptic input and channel current as the postsynaptic output. **b** Transfer curves of OECTs inoculated with *S. oneidensis* MR-1. **c** Illustrations of the paired pulse input from the gate (**c**, upper plot), and the corresponding channel current changes $\triangle I_{DS}$ of biotic and abiotic OECTs (**c**, lower plot). **d** The channel conductance changes $\triangle G$ plotted with varying pulse voltage $V_P$, while pulse duration $t_P$ and pulse interval $\triangle t$ were fixed at 80 ms. **e** Channel conductance changes $\triangle G$ for varying $\triangle t$, with fixed $t_P = 80$ ms, $V_{DS} = -0.05$ V, and $V_P = 0.5$ V. **f** Continuous voltage pulses were applied to the gates (**f**, upper plot) and the corresponding channel conductance for *S. oneidensis* MR-1 inoculated OECTs (**f**, lower plot). **g** Conductance changes $\triangle G$ for

OECTs inoculated with $\triangle$Mtr knockout strains complemented with either *MtrCAB* Buffer gate or empty vector plasmids. $\triangle G$ *p* values for pairs indicated from top to bottom $p = 0.0115$ and $p = 0.0046$. The $\triangle G$ corresponds to $\triangle mtrC$ mutants carrying **h** NAND and **i** NOR Boolean logic gates with different inducer combinations. Contrast tests for the $\triangle G$ showed *p* values for **h** $p = 0.00306$ and panel **i** $p = 0.00152$. Data show the mean ± SD of 3 biological replicates. For statical analysis, unpaired two-tailed Student's t-tests were performed without adjustments for multiple comparisons in **g** where n.s. represents $p > 0.05$. General linear hypothesis tests were used to evaluate interaction terms and linear contrasts for logic 1 s and logic 0 s of the NAND and NOR logic gates in **h** and **i**. **a** created with BioRender.com.

$\triangle$Mtr +*mtrCAB*) showed similar conductance changes and A2/A1 index values to those of *S. oneidensis* MR-1, while negative controls ($\triangle mtrC$ +empty and $\triangle$Mtr +empty) behaved similarly to abiotic OECTs (Fig. 5g, Fig. S5e). These data suggest that the synaptic functions in biotic OECTs are directly correlated with EET activity from the Mtr pathway. To further examine the extent of genetic control over

synaptic behavior, the $\triangle mtrC$ strains carrying Boolean logic gates (NAND and NOR) were subjected to different inducer combinations. As shown in Fig. 5h, i, strains expressing logic output 1 had marked weight changes compared to the logic output 0 strains, demonstrating computational control over synaptic function in response to specific environmental cues. Although the detailed mechanisms underlying

the correlation between synaptic behavior and bacterial EET are not yet clear, our current findings establish a direct link between the programmable synaptic response and cellular EET. By demonstrating the direct involvement of EET in shaping synaptic behavior, these results pave the way for further studies on artificial synapses and biocomputing.

## Discussion

OECTs are powerful devices for interfacing with biological systems because they inherently couple ionic and electronic signaling. While the majority of OECT applications leverage biological changes to alter ion transport or induce electrochemical changes in the channel, we demonstrated that biological electron transport from living bacteria can also be coupled to OECT sensing and computation. Specifically, we found that the model electroactive bacterium *S. oneidensis* MR-1 could change the conductivity and doping state of the PEDOT:PSS channel via EET. This process was directly tied to the presence of metabolically active cells. By controlling the bias potential, the kinetics of electron transfer between cells and electrodes could be modulated, providing a means to tune electron transfer to the channel[48]. During our OECT operation, the use of a polarizable Au gate instead of a non-polarizable Ag/AgCl gate could introduce inaccuracies when measuring the channel potential due to capacitive effects on the Au gate and uncertainties in the onset of redox reactions on the gate and channel due to the lack a fixed electrode potential[49,50]. Consequently, to enable precise measurements of the source potential, we used Ag/AgCl pellet pseudo-reference electrodes to investigate the de-doping mechanisms. Combined with UV-Vis spectroscopy and alternative OECT designs, we showed that *S. oneidensis* can interact with both the gate and the channel, resulting in charge accumulation and change in the doping state of the channel. Furthermore, we established that the gate bias voltages could be leveraged to tune electrode potential, consequently regulating EET to the electrode and channel. This gate-voltage-controlled modulation of cellular EET de-doping efficacy underscores the intricate interplay between biological and electronic components in the hybrid OECTs. However, the electron transfer and the de-doping via the cell-channel interface (path 1 in Fig. 2b) can still mingle with the cell-gate path unless extreme bias potentials are used, which could result in undesirable background electrochemical reactions and stress on bacteria cells. Thus, when investigating gate-induced doping with EET (path 2 in Fig. 2b), it may be advantageous to only allow EET between the cell and gate electrode while segregating the channel from any EET activities. Although we did not isolate bacteria to one part of the device, future OECTs could accomplish this using light-patternable biofilms[51], ion-permeable membranes to separate the gate and channel, or multiple gates[52]. Alternative device architectures, such as the use of floating gates, could also be employed to enhance amplification from EET[50,53].

On the timescale of our experiments, direct EET through the Mtr pathway dominated, but flavins also accelerated channel de-doping when added exogenously. Biofilm formation was also not required to observe the desired response result, consistent with previous studies on OECTs containing *S. oneidensis*[18]. This is promising because genetic circuits typically function best in planktonic, actively growing bacteria[54]. The diminished role of flavins during normal growth in our OECTs is in contrast to previous studies of *S. oneidensis*-PEDOT:PSS composite electrodes, which found that flavins were critical for current generation[55]. These differences can likely be explained by different surface morphology, hydrophobicity, and other chemical factors affecting the conducting polymer in the bacteria-PEDOT:PSS composite electrode, as compared to the PEDOT:PSS film utilized in our work. Indeed, post-processing of PEDOT:PSS can drastically alter its conductivity, surface properties, and redox potential[56]; these factors likely dictate polymer-bacteria interactions and warrant more systematic investigation. Finally, a variety of other (semi)conducting polymers

have been evaluated in OECTs. Future research aimed at exploring the interaction of *S. oneidensis* and new materials[57], especially n-type conductive polymers, is likely to result in OECTs with enhanced sensitivity, response time, operation efficiency, and other desirable characteristics.

Relative to traditional microelectronics, living systems offer several computational advantages such as enhanced efficiency, self-repair, and the inherent capability for parallel, distributed, and adaptive computation. These features make living systems well-suited for various applications, including biosensing and novel computational paradigms like neuromorphic computing, amorphous computation, and morphological computation[58]. To showcase the potential advantages of electroactive bacteria-inoculated OECTs in biosensing and biocomputing, we utilized plasmid-based Boolean logic gates to control cellular EET flux, achieving the direct conversion of transcriptional logic into electrical signals in response to combinations of chemical stimuli. Our modular genetic circuits regulated the expression of different parts of the Mtr pathway, since we demonstrated that these proteins directly contribute to changes in channel conductivity. However, enzymes and proteins upstream of this pathway in *S. oneidensis* or other Mtr homolog-containing bacteria (*Aeromonas hydrophillia*, *Vibrio natrigens*, etc.) may be enticing targets for future engineering since metabolic flux is directly connected to EET[59,60]. We chose transcriptional regulation for our circuits, but in principle any type of genetic regulation could be connected to an OECT output. For example, ferredoxin circuits[24], integrases[14,61], anti-repressors[62], and other genetic regulatory motifs could be used to optimize signal transduction, dynamic range, and gate simplicity[63]. Overall, our system demonstrates the translation of Boolean logical computations in *S. oneidensis* to electrical readouts. Within this EET framework, we envision OECTs as a foundational platform for connecting practically any metabolic, genetic, or protein-based circuit to an electronic output.

Lastly, as a demonstration of future biocomputing applications, we employed OECTs containing electroactive bacteria as a platform to emulate synaptic behavior. Leveraging inducible control over EET flux, we established a correlation between EET and synaptic modulation. Biotic OECTs exhibited distinct paired pulse responses and conductance changes compared to abiotic devices. With positive gate pulses, the channel exhibited a notable spike-recovery response to both the rising and falling edge of the gate pulse (Fig. S11a), resulting in a short-term increase in the conductance and doping of the channel. The spike-recovery response is likely due to the relatively low $V_{DS}$ ($-0.05\,V$) compared to $V_{GS}$ (up to 0.5 V) where the transport of ions from the electrolyte into the channel outpaces the hole extraction rate within the channel[8]. However, when applying negative gate pulses, the channel exhibited a step-like response and little short-term conductance change (Fig. S11b). Although the cause of the observed asymmetric modulation and spike-recovery behavior remains unclear, comparing the results between strains with different EET protein expression levels, it is evident that the channel response arose from cellular EET, and cannot be explained by the blockade of ionic movement from the presence of cells alone. Figure S11c illustrates a distinct behavior observed in the drain electrode potential at the onset of positive gate pulse ($V_P = 0.5\,V$), wherein a sharp spike occurred below $-0.9\,V$ (vs Ag/AgCl). Conversely, as shown in Fig. S11d, during the negative gate pulses ($V_P = -0.5\,V$) the drain potential increased beyond $-0.3\,V$ (vs Ag/AgCl). Given that the drain-source voltage was consistently maintained at $-0.05\,V$, it is reasonable to assume that the channel experienced a similar potential range during the gate pulses. Consequently, we hypothesize that the channel's low potential around $-0.9\,V$ might impede or disrupt electron transfer between the channel and attached cells, resulting in the observed asymmetric spike-recovery response to positive gate pulses. The long-term plasticity of the biotic OECTs is mainly

dependent on cellular EET, as cells de-dope the channel, the conductance decreases accordingly. To determine whether EET de-doping induced changes in hole concentration and extraction rate would cause short-term memory effect due to pronounced spike-and-recovery transient response[8], we continuously biased the source electrode at varying potentials against the Ag/AgCl RE in abiotic OECTs, emulating EET de-doping. Our results (Fig. S12a) reveal that under different channel de-doping states, induced by paired pulses of 0.5 V across the gate and source, there were no short-term conductance increases similar to those in biotic OECTs. $I_{DS}$ responses at source bias potentials of −0.5 V and −0.6 V were normalized to initial values for clearer visualization (Fig. S12b), reflecting the typical $I_{DS}$ reduction percentages (75.8% and 87.3%, respectively) associated with EET de-doping. These findings suggest that the long-term de-doping process induced by EET does not contribute to a short-term memory effect. To better control long-term plasticity in OECTs and potentially reset synaptic weight decrease caused by bacterial EET, oxygen was introduced to facilitate synaptic weight increase. As depicted in Fig. S13a, transitioning from the EET de-doped anaerobic state to aerobic conditions with a 3-h exposure to ambient oxygen resulted in a 75% recovery in the channel current. Additionally, a notable recovery in the measured source potential ($V_S$, vs Ag/AgCl) was observed (Fig. S13b–d). Further work is needed to unravel the mechanism of EET incurred spike-recovery channel response and improve the practicality of the biotic OECTs as artificial synapses. For example, reference electrodes can be used to apply a bias voltage to the channel, allowing the channel potential to be precisely controlled or monitored to study its dynamics. Additionally, these reference or auxiliary electrodes offer a convenient means of resetting the OECT in artificial synapse applications by independently controlling the doping state of the channel. New channels utilizing n-type or inorganic semiconducting materials could also provide valuable insights into the redox behaviors of the electroactive bacteria and expand device designs to enable more efficient synapses or complementary OECTs[13,64]. Similarly, new biopolymers and cell-derived materials, such as light-sensitive electronic-protonic conductors[65], microbial nanowires[66], and cell-secreted dopamine[67] portend an ongoing fusion of materials, living cells, and electronics. Ultimately, the ability of EET to change synaptic weight in response to genetic memory, transcriptional regulation, and other forms of biological computation will enable new forms of metaplasticity in OECTs[68].

OECTs have emerged as an ideal platform for combining biological and electrical signaling. Despite the more accessible experimental tools for rapid genetic and metabolic manipulation of bacteria, the existing body of research connecting bacterial cellular processes with OECTs remains limited. Our study demonstrates the viability of employing OECTs to translate bacterial computation that modulates the extracellular redox environment via extracellular electron transfer. We note that in addition to the transcriptional Boolean logic gates, analog biocomputing schemes are also compatible with the OECT platform and associated future system designs. The inclusion of living cells in our hybrid devices imparts unique bio-mimetic properties such as self-regulation, biological sensing and computation, and short-term synaptic memory. Overall, our work integrates knowledge and techniques from bioelectronics, synthetic biology, and electrochemistry to create a versatile platform for future biosensing and biocomputing systems.

## Methods

For detailed descriptions of the methods used, please refer to the Supplementary Information.

### Reporting summary

Further information on research design is available in the Nature Portfolio Reporting Summary linked to this article.

## Data availability

Experimental data supporting the findings of this study are available through the Texas Data Repository (https://doi.org/10.18738/T8/MNKO8D).

## Code availability

The codes supporting the findings of this study are available in the Texas Data Repository (https://doi.org/10.18738/T8/MNKO8D).

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

## Acknowledgements

Base plasmids for the NAND circuit were generously provided by the Voigt Lab via Addgene (#49375, #49376, #49377). This research was financially supported by the Welch Foundation (Grant F-1929, B.K.K.), the National Institutes of Health under award number R35GM133640 (B.K.K.), an NSF CAREER award (1944334, B.K.K.), and the Air Force Office of Scientific Research under award number FA9550-20-1-0088 (B.K.K.). A.J.G. was supported through a National Science Foundation Graduate Research Fellowships (Program Award No. DGE-1610403). AFM experiments were performed on an instrument obtained through an AFOSR DURIP award (FA9550-21-1-0148). The authors acknowledge use of shared research facilities supported in part by the Texas Materials Institute, the Center for Dynamics and Control of Materials: an NSF MRSEC (DMR-1720595), and the NSF National Nanotechnology Coordinated Infrastructure (ECCS-1542159). We gratefully acknowledge the use of facilities within the core microscopy lab of the Institute for Cellular and Molecular Biology, University of Texas at Austin. Cartoon illustrations were created using BioRender.com. We acknowledge A.J.G. for his contribution to Fig. S10, originally featured in Transcriptional Regulation of Synthetic Polymer Networks, https://doi.org/10.1101/2021.10.17.464678, and reproduced here with permission.

## Author contributions

Y.G. and B.K.K. conceived the project and designed research. Y.G. performed the majority of experiments and analysis with device fabrication assistance from Y.Z., B.T. and A.D. X.J. and J.R. assisted with device characterization and analysis. A.J.G., C.M.D., I.E.M.M. and B.M.T. constructed and characterized genetic circuits. G.P. performed the statistical analysis. Y.G. and B.K.K. wrote the manuscript with input and assistance from all authors.

## Competing interests

The authors declare no competing interests.
