## [Peer Review File · Nature Communications]

REVIEWER COMMENTS

Reviewer #1 (Remarks to the Author):

The article entitled, " A Hybrid Transistor with Transcriptionally Controlled Computation and Plasticity" with the manuscript ID NCOMMS-23-44703-T, aimed to demonstrate extracellular electron transfer (EET) in *S. oneidensis* using organic electrochemical transistor (OECT) and EET-driven changes to synaptic plasticity of the device. While the author aimed to give a different perspective of EET using an OECT device, the proposed method/mechanisms lack evidence and statistical analysis and significantly mislead the EET phenomena in biological medium. Overall, I do not think that the manuscript displays a high enough level of accuracy to demonstrate the feasibility of this concept. Besides many shortcomings and inaccuracies in EET mechanisms, it is not clear why Authors also considered " Transcriptionally Controlled Computation". How does this part of the study help the general understanding of EET or any biological study? This part of the paper is also misleading. There is no computation or anything similar described in the paper. The Authors only categorized the signal that comes from different strains into Boolean Logic operations, which can be any electrode material. One does not need any "synaptic plasticity" of the OECT device. The paper sounded more like a combination of popularly known subjects in one platform, therefore it does not match the quality standards to be published in an important journal like Nature Communication. However, it is highly recommended to consider the below points before submitting the paper elsewhere.

- Before re-considering the reviewing the paper, it is highly recommended for Authors to read EET phenomena in similar electrochemical systems (Nature volume 491, pages218–221 (2012), Nature Nanotechnology volume 11, pages910–913 (2016), J. Am. Chem. Soc. 2018, 140, 32, 10085–10089). This paper may give a clear idea of how electron transfer occurs in biological media, and how it transfers to the electrode surface.

- The authors explained the electron-transfer mechanisms to the channel electrode coated with a polymer and schematically demonstrated in Figure 2b. Firstly, how do the Authors think that the electron "jumps" from the solution to the PEDOT:PSS surface? Is there any electron-transfer mediator to help the electron transfer from Flavin to the PEDOT:PSS?

- As described in mechanisms, the Authors only show electron transfer that changes the conductivity of the channel, however, they also used a growth medium that contains many ions, therefore it is natural that they also diffuse into the PEDOT:PSS and change the conductivity. There is no explanation for the contribution of ion diffusion. Why did the Authors ignore the ion diffusion?

- Authors used Ag/AgCl as gate electrode as described in supporting information. However, it is well-known that Ag/AgCl is bactericidal. How does this affect the growth of bacterial culture? Is there any control study?

- All data were presented as data points and histograms in all figures, which significantly reduces the reliability of the data presented. How were these transfer curves vs time data obtained?

- The Figure 2f is quite confusing. Apparently, the device also shows response in the absence of bacteria in the medium even faster changes of drain current upon gate voltage changes. How do Authors explain the similar trends obtained in Figure 2f?

Reviewer #2 (Remarks to the Author):

The manuscript by Gao et al. describes the use of bacterium in combination with OECT's to develop several sensing computation and synaptic functions. Overall the results are interesting and well-described. At the start of the manuscript it reads a bit like a chronological story of what was done, including a list of functionality that the device could achieve. Furthermore, it is certainly a long paper with a lot of details. However, I do believe the work presented is unique and fits well in this journal. I have a few comments:

1. I believe the first paragraphs of the results can be shortened and the main fabrication steps should be included in the methods, not the main text.

2. The statement "A crucial property of synaptic transistors is non-volatility, which can be visualized as hysteresis in the transfer curve." is only partly true. Hysteresis results from the timescale difference in ionic and electronic charges, as ions are still flowing out (or in) when one is already back at the same voltage while measuring the IV curve. However, this effect is generally attributed only to short-term plasticity, with a certain timescale in the order of seconds/minutes. Long-term plasticity is assumed to be longer than that and requires some form of trapping charges (in the case of OECTs), either by structural trapping or electrical charge trapping (open circuits), or by some form of redox reaction doping the polymer. The PPF in Figure 5c is indeed way of demonstrating the "slow kinetics" effect of short term plasticity, but in this also highlights the residual non-volatile effect of electron transfer/doping. ^{[1][2][3][4]}_{[5][6][7][8]} The authors should rephrase the statement, while discussing how the long-term plasticity effect is a result of redox reactions that dope/dedope the mixed conductor (which is apparent in Figure 5c bottom graph and visualized in Figure 2b)

3. Related to that, is it also possible to return to the original level of conductance (something that is important for synaptic weights in neural networks)? Generally this can be done with ambient (or dissolved) oxygen in the electrolyte, but this is not stated somewhere. Perhaps include a short discussion on this too.

Reviewer #3 (Remarks to the Author):

Manuscript, extended data, and SI are extremely thorough and well written. Intricacies of making OECT measurements are outside of my expertise, but the approach the authors use for incorporating electroactive bacteria into the system is very rigorous and includes all proper controls. Therefore my review only contains a few technical questions and focuses more on how the authors might reinforce the significance of their findings wrt microbial electrochemical technologies. Authors lay out rationale for OECT R&D for bioelectronics but I do not yet see a connection for why it is important to develop OECT for microbial electronics. The novelty, and even benefit, of utilizing a OECT to study microbial electron transfer, particularly for genetically engineered strains is clear despite the system being potentially quite complicated. However, I do not see the connection between electroactive bacteria and application of OECT, i.e. is there a reason to connect electroactive bacteria to OECT other than studying the bacteria? While I realize this is a basic research paper, I am looking for some rationale as to why a system as complex as an OECT would be beneficial for sensing in a system where microorganisms are present. I may have missed an explicit statement about this benefit, but what I would like to see is something that indicates a fold increase in sensitivity or other over what can currently be achieved with much simpler systems. I am not asking that the authors show this experimentally, but there needs to be a more convincing argument, in my mind, as to why OECT are helpful for microbial electrochemical technologies.

examine material stability within the device, OECTs

171 containing *S. oneidensis* were gently washed with soapy water and examined with atomic force

172 microscopy (AFM).

Why is soapy water used? Does this not result in cell lysis?

Was the longest experiment 25 hours? Wondering about PEDOT/PSS stability or a longer period of time.

Prior to inoculation into OECTs, all strains

316 were anaerobically induced for 18–24 hours with 1mM IPTG or 100 nM OC6 to ensure steady317

state protein expression.

Were these strains then inoculated in a non-growth state and presumed that cells interacting with the OECT already had EET proteins expressed? Was the inducer included in the OECT chamber? Please clarify

How does timing/growth in the presence of the inducer affect the Figure 4a and b? Presumably protein expression is limited by the amount of inducer, but I do not see that data shown anywhere by measuring protein concentration or by measuring a different marker, like fluorescence. Was this published previously?

NCOMMS-23-44703-T

Original reviewers' comments are in black, responses are marked in blue.

Reviewer #1:

The article entitled, " A Hybrid Transistor with Transcriptionally Controlled Computation and Plasticity" with the manuscript ID NCOMMS-23-44703-T, aimed to demonstrate extracellular electron transfer (EET) in *S. oneidensis* using organic electrochemical transistor (OECT) and EET-driven changes to synaptic plasticity of the device.

We thank the reviewer for taking the time to review this manuscript.

While the author aimed to give a different perspective of EET using an OECT device, the proposed method/mechanisms lack evidence and statistical analysis and significantly mislead the EET phenomena in biological medium. Overall, I do not think that the manuscript displays a high enough level of accuracy to demonstrate the feasibility of this concept. Besides many shortcomings and inaccuracies in EET mechanisms, it is not clear why Authors also considered " Transcriptionally Controlled Computation". How does this part of the study help the general understanding of EET or any biological study?

Thank you for your valuable feedback and for raising an important question regarding the inclusion of transcriptionally controlled computation in our study. Transcriptionally controlled computation allows us to use gene expression as a biological/electrochemical readout for information processing within microbial systems. This approach can be instrumental in the study of EET for several reasons:

1. Studying regulatory networks. We can dissect and understand the complex regulatory networks that control EET or other genes of interest. This can also shed light on the natural decision-making processes that microorganisms rely on to control electron transfer in response to complex environmental cues.

2. Biocomputation and biotechnologies. With the genetic circuits controlling when and how EET genes are activated in response to environmental stimuli, the circuits can serve as models and building blocks to create complex and responsive systems. For example, it is possible to screen for environmental or chemical factors by observing changes in gene expression patterns, or bacteria can be engineered to initiate EET-base biodegradation only in the presence of specific contaminants/conditions. While our studies do not address the general or fundamental understanding of EET biology, they do highlight how EET can serve as a unique interface between biological and microelectronic devices.

This part of the paper is also misleading. There is no computation or anything similar described in the paper. The Authors only categorized the signal that comes from different strains into Boolean Logic operations, which can be any electrode material.

In our paper, we developed synthetic genetic circuits that function as Boolean logic gates within bacterial cells to activate EET gene expression in response to specific combinations of inducer molecules. When we refer to 'computation', we mean the cellular decision-making process that occurs as a result of interpreting multiple environmental signals through the genetic circuits. We note that the computations performed by the bacteria via these genetic circuits are not binary in the same sense as semiconductor-based counterparts. Biological systems often exhibit a range

of responses rather than a strict 'on' or 'off' state. Therefore, the logic gates we describe should be understood as regulatory mechanisms that allow for a gradation of gene expression levels in response to varying concentrations of inducers. Consequently, by categorizing the bacterial responses to these inducers using the framework of Boolean logic, we are utilizing this framework as a conceptual tool to understand and predict the behavior of our genetic circuits and building blocks for further complex logic. For instance, in mixed cultures one species could be engineered to perform EET only when another species provides certain signals, establishing a controllable symbiotic relationship. These responses, while influenced by the concentrations of inducers, ultimately guide the bacterial decision to engage in EET, thus representing a form of biological computation.

It is true that other bioelectrochemical systems could potentially be used with the same strains. However, we demonstrate that OECTs are uniquely positioned for our application due to their small size, high transconductance, and other factors. To further address this topic, we added more discussion comparing OECTs and traditional bioelectrochemical systems (Lines 96-109) in response to Referee 1 and related comments from Referee 3.

One does not need any ‘‘synaptic plasticity’’ of the OECT device.

The field of OECTs is still emerging, but there is significant interest in OECT-based artificial synapses. While contributions to this area are currently in a nascent stage, the synaptic plasticity and neuromorphic computing within the OECT framework is increasingly recognized in the fields of bioelectronics and biocomputation. Please see the works below for examples:

- van de Burgt, Y. et al. A non-volatile organic electrochemical device as a low-voltage artificial synapse for neuromorphic computing. *Nat. Mater.* 16, 414–418 (2017).
- Chen, Shuai, et al. Recent technological advances in fabrication and application of organic electrochemical transistors. *Advanced Materials Technologies* 5, 12, 2000523 (2020).
- Yan, Y. et al. High-performance organic electrochemical transistors with nanoscale channel length and their application to artificial synapse. *ACS Appl. Mater. Interfaces* 12, 49915–49925 (2020).
- Ji, Xudong, et al. Mimicking associative learning using an ion-trapping non-volatile synaptic organic electrochemical transistor. *Nature communications* 12, 1, 2480 (2021).
- Liu, G. et al. Ultralow-Power and multisensory artificial synapse based on electrolyte-gated vertical organic transistors. *Adv. Funct. Mater.* 32, 2200959 (2022).

Of course, plasticity is not required for every application. However, in our case, it is an excellent demonstration of how engineering of the bacteria and device can lead to a synergistic outcome that is challenging to realize with either component individually.

The paper sounded more like a combination of popularly known subjects in one platform, therefore it does not match the quality standards to be published in an important journal like *Nature Communication*. However, it is highly recommended to consider the below points before submitting the paper elsewhere.

- Before re-considering the reviewing the paper, it is highly recommended for Authors to read EET phenomena in similar electrochemical systems (*Nature* volume 491, pages218–221 (2012), *Nature Nanotechnology* volume 11, pages910–913 (2016), *J. Am. Chem. Soc.* 2018, 140, 32,

10085–10089). This paper may give a clear idea of how electron transfer occurs in biological media, and how it transfers to the electrode surface.

We are very familiar with these papers. Figures 1-3 as well as Extended Figures 1 and 2 are devoted to unraveling how bacteria interface electronically with the OECT.

- The authors explained the electron-transfer mechanisms to the channel electrode coated with a polymer and schematically demonstrated in Figure 2b. Firstly, how do the Authors think that the electron “jumps” from the solution to the PEDOT:PSS surface? Is there any electron-transfer mediator to help the electron transfer from Flavin to the PEDOT:PSS?

During the 24-hour test, we believe that electrons were transferred to the electrode and the PEDOT:PSS channel through a direct contact pathway. This is substantiated by the findings in Figure 3b, where the Δbfe strain (a mutant with diminished flavin secretion) exhibited a similar I_{DS} reduction rate to that of the wild-type strain (MR-1). Secondly, flavins do indeed play a pivotal role in enhancing electron transfer to both the electrode and the PEDOT:PSS channel. When exogenous FMN was introduced, there was a significant increase in the I_{DS} reduction rate for both the wild-type strain and strains impaired in flavin secretion. This observation is further detailed in Figure 3b. Furthermore, as illustrated in Figure 3c, a sigmoidal relationship was observed between the concentration of exogenous FMN and the I_{DS} reduction rate, which is consistent with the direct EET transfer mechanism via FMN-bound MtrC. Finally, as highlighted in the manuscripts mentioned above, as well as a significant body of work on *S. oneidensis*, the Mtr pathway is also critical for electron transfer.

We acknowledge that we do not have a full molecular understanding of electron transfer between protein and materials, but this is not unique to PEDOT:PSS and is a limitation of the field in general.

- As described in mechanisms, the Authors only show electron transfer that changes the conductivity of the channel, however, they also used a growth medium that contains many ions, therefore it is natural that they also diffuse into the PEDOT:PSS and change the conductivity. There is no explanation for the contribution of ion diffusion. Why did the Authors ignore the ion diffusion?

In addressing the potential contribution of ion diffusion, we employed abiotic controls, which lacked the bacterial cells but were subjected to identical electrode bias voltages as the biotic samples and tested alongside biotic OECTs. Both abiotic and biotic devices did show typical OECT behavior (Figure 2f); voltage sweeps were associated with ions (from the growth medium) moving in and out of the PEDOT:PSS channel. We also comment on the general role of ion diffusion and OECT operation in Lines 205-210 and do not believe these points need further elaboration as our focus is on how *S. oneidensis* influences OECT operation.

Specifically, as depicted in Figure 2c, 2f, and Extended Data Figures 2b-2f, the abiotic samples exhibited negligible I_{DS} decay. Moreover, taking the ionic change due to cell growth, the supernatants containing ions and metabolic products from cell growth were also inoculated into the OECT and no obvious current changes were observed (Extended Data Figures 2e and 2f). Together, these results suggest that the observed I_{DS} reduction (Figure 2c) is predominantly attributable to the metabolically active *S. oneidensis* cells, rather than ion diffusion from the growth medium into the PEDOT:PSS.

- Authors used Ag/AgCl as gate electrode as described in supporting information. However, it is well-known that Ag/AgCl is bactericidal. How does this affect the growth of bacterial culture? Is there any control study?

We apologize for the confusion surrounding the Ag/AgCl electrodes and note that the Ag/AgCl pseudo-reference electrodes were transiently introduced into the OECT chambers solely during transfer curve measurements, which lasted for just a few minutes per run, as depicted in Figure 4c-4f, Extended Data Figure 3a, and Extended Data Figures 4c and 4d. They were not present for the vast majority of experiments. Finally, in experiments where the Ag/AgCl electrodes were consistently present for an extended duration (~13 hours), observable in Extended Data Figure 3b, the I_{DS} still exhibited the anticipated decrease, similar to OECTs that were devoid of the Ag/AgCl electrodes. This indicates a minimal antibacterial influence exerted by the Ag/AgCl electrodes.

- All data were presented as data points and histograms in all figures, which significantly reduces the reliability of the data presented.

We are confused by this comment and wonder about alternative means to present the collected data. For clarity, the heat maps in Figure 4a and 4b depict the averages of biological triplicates, without the inclusion of error bars. We apologize for the error but have now included the raw I_{DS} curves for these experiments that show averages and standard deviations for each condition (Figure S4).

The UV-Vis spectra shown in Figure 2g, 5b, Extended Data Figure 3d, and 3e, as well as the long-term measurement in Figure 5f and Extended Data Figure 3b, were derived from individual devices. However, the rest of our figures present data as the mean and standard deviation of biological triplicates. Error bars are indicated by either distinct lines or shaded areas. Some error bars were not observable due to their small dimensions in comparison to the data symbol, which underscores the high consistency across our replicates. Moreover, all datasets underwent rigorous statistical analysis to maintain the credibility and robustness of our results. We note that the number of experimental replicates and statistical analysis is well beyond what is typical for microbial fuel cell or other bioelectrochemical papers.

- How were these transfer curves vs time data obtained? The Figure 2f is quite confusing. Apparently, the device also shows response in the absence of bacteria in the medium even faster changes of drain current upon gate voltage changes. How do Authors explain the similar trends obtained in Figure 2f?

As mentioned in Line 221, the transfer curves were measured 24 hours after the inoculation of the OECT with MR-1 cells. For the abiotic controls, no inoculations were carried out, and they were exposed to the same bias voltages and durations as the biotic ones.

For transfer curve characterization, generally, for p-type OECTs, like the PEDOT:PSS channel in our experiment, it's typical to observe the channel turning-off (decreasing I_{DS}) with increased gate bias voltages (V_{GS}). This is the expected result. Comparing the MR-1 and abiotic curves, a couple of observations are worth highlighting:

- A pronounced reduction in the I_{DS} for MR-1 samples was evident, pointing to a decrease of channel conductance with equivalent applied V_{GS} .
- Taking into account the effective gate potentials, denoted on the x-axis in Extended Data Figure 3a, the MR-1 sample transfer curves exhibited a shift towards more positive

effective gate potential (or more negative source potential). These shifts indicate the reduction of the electrodes and channel turning-off behavior caused by the MR-1 cells.

Reviewer #2:

The manuscript by Gao et al. describes the use of bacterium in combination with OECT's to develop several sensing computation and synaptic functions. Overall the results are interesting and well-described. At the start of the manuscript it reads a bit like a chronological story of what was done, including a list of functionality that the device could achieve. Furthermore, it is certainly a long paper with a lot of details. However, I do believe the work presented is unique and fits well in this journal. I have a few comments:

We thank the reviewer for taking the time to review this manuscript.

1. I believe the first paragraphs of the results can be shortened and the main fabrication steps should be included in the methods, not the main text.

Thank you for the suggestion. We have condensed the first paragraphs of the result section and moved the fabrication details to the method section to maintain the focus of the main text.

2. The statement “A crucial property of synaptic transistors is non-volatility, which can be visualized as hysteresis in the transfer curve.” is only partly true. Hysteresis results from the timescale difference in ionic and electronic charges, as ions are still flowing out (or in) when one is already back at the same voltage while measuring the IV curve. However, this effect is generally attributed only to short-term plasticity, with a certain timescale in the order of seconds/minutes. Long-term plasticity is assumed to be longer than that and requires some form of trapping charges (in the case of OECTs), either by structural trapping or electrical charge trapping (open circuits), or by some form of redox reaction doping the polymer. The PPF in Figure 5c is indeed way of demonstrating the “slow kinetics” effect of short term plasticity, but in this also highlights the residual non-volatile effect of electron transfer/doping. The authors should rephrase the statement, while discussing how the long-term plasticity effect is a result of redox reactions that dope/dedope the mixed conductor (which is apparent in Figure 5c bottom graph and visualized in Figure 2b)

We thank the reviewer for their comments and have implemented their suggested changes (Lines 399-401). We primarily evaluated short-term plasticity after the decrease in channel current had reached saturation (about 20 hours). This allowed us to isolate short-term plasticity effects without having to worry about longer time scale phenomena. However, the reviewer is correct that the bacteria-induced de-doping of the channel can be thought of as a long-term plasticity change that also depends on EET. We have modified the discussion (Lines 573-583 and SI Figure S7 to emphasize this point).

3. Related to that, is it also possible to return to the original level of conductance (something that is important for synaptic weights in neural networks)? Generally this can be done with ambient (or dissolved) oxygen in the electrolyte, but this is not stated somewhere. Perhaps include a short discussion on this too.

Yes, this is possible as shown in Extended Data Figure 2b. We also conducted additional experiments following short-term plasticity voltage pulsing to show that the channel conductance returns to the original level following ambient oxygen exposure (Figure S8).

We have not repeatedly cycled between anaerobic and aerobic but are working on electrochemical methods to generate oxygen in situ to make resetting synaptic weights easier. Interestingly, the bacteria will also 'reset' in the presence of oxygen, as they prefer it as an electron acceptor. Per the reviewer's suggestion, we have included additional discussion on these topics (Lines 583-588).

Reviewer #3 (Remarks to the Author):

Manuscript, extended data, and SI are extremely thorough and well written. Intricacies of making OECT measurements are outside of my expertise, but the approach the authors use for incorporating electroactive bacteria into the system is very rigorous and includes all proper controls. Therefore my review only contains a few technical questions and focuses more on how the authors might reinforce the significance of their findings wrt microbial electrochemical technologies.

We thank the reviewer for taking the time to review this manuscript.

Authors lay out rationale for OECT R&D for bioelectronics but I do not yet see a connection for why it is important to develop OECT for microbial electronics. The novelty, and even benefit, of utilizing a OECT to study microbial electron transfer, particularly for genetically engineered strains is clear despite the system being potentially quite complicated. However, I do not see the connection between electroactive bacteria and application of OECT, i.e. is there a reason to connect electroactive bacteria to OECT other than studying the bacteria? While I realize this is a basic research paper, I am looking for some rationale as to why a system as complex as an OECT would be beneficial for sensing in a system where microorganisms are present. I may have missed an explicit statement about this benefit, but what I would like to see is something that indicates a fold increase in sensitivity or other over what can currently be achieved with much simpler systems. I am not asking that the authors show this experimentally, but there needs to be a more convincing argument, in my mind, as to why OECT are helpful for microbial electrochemical technologies.

We thank the reviewer for their comment. The rationale for integrating electroactive bacteria with OECT is twofold. Firstly, as the reviewer stated, the OECTs can act as sensors for bacterial EET. Secondly, the electroactive bacteria enrich the functions of OECTs. For instance, bacteria engineered with transcriptionally controlled logic gates can modulate OECT outputs based on the sensing and computation performed by resident cells, meaning that the hybrid transistors can provide readouts responding to specific environmental conditions. Similarly, the EET has the potential to influence the synaptic weight of the OECT-based artificial synapses, adding a biologically-driven component to the electronic information processing. Together, the bacteria can be engineered to allow OECTs to respond to new stimuli, such as light or specific chemicals, broadening their utility for biosensing applications.

Although improving sensitivity was not our primary objective for the hybrid transistor, the inherent amplification capability of OECT can still be observed in our results. Traditional microelectrodes typically record bacterial EET current in the nA range, as can be seen from the I_{GS} in Figure 3f, and other published works as J. Am. Chem. Soc. 2018, 140, 32, 10085–10089. This is mostly

due to the size of the electrode, the number of bacteria cells, and biofilm formation. Contrarily, in a comparable setup, the channel current I_{DS} from OECTs is usually in μA range, as can be seen in Figure 2f and Extended Data Figure 3b. Thus, there is around a 1000-fold increase in the detection current by the OECTs. We added a short discussion (Lines 99-105) to emphasize this point.

1. examine material stability within the device, OECTs

171 containing *S. oneidensis* were gently washed with soapy water and examined with atomic force

172 microscopy (AFM).

Why is soapy water used? Does this not result in cell lysis?

In this case, we were specifically examining whether the presence of *S. oneidensis* damages the PEDOT:PSS film over time via delamination, pitting, or other effects. Thus, after incubating with the device, live cells were no longer required.

Was the longest experiment 25 hours? Wondering about PEDOT/PSS stability or a longer period of time.

Devices are stable for longer periods of time (up to 96 hours, Extended Data Figure 2a and 2b). Under anaerobic conditions, the changes in current and PEDOT:PSS conductivity remain, even after cells are no longer metabolically active (effectively the cells have fully de-doped the channel). We also showed that devices are stable during cycling between anaerobic and aerobic conditions, which re-oxidize the PEDOT:PSS and essentially resets the device (Extended Figure 2b, new Figure S8). We did not explore how many times we could cycle the device before failure, but this is a subject of ongoing work.

Prior to inoculation into OECTs, all strains

316 were anaerobically induced for 18–24 hours with 1mM IPTG or 100 nM OC6 to ensure steady 317state protein expression.

Were these strains then inoculated in a non-growth state and presumed that cells interacting with the OECT already had EET proteins expressed? Was the inducer included in the OECT chamber? Please clarify

Yes, the strains were inoculated into the OECT chambers in a non-growth state, with the assumption that they had already expressed the relevant EET proteins. Inducers were also present in the OECT chamber, maintained at the same concentrations used during the induction phase. We have modified the manuscript (Lines 353-355) and updated the 'Methods/Device Operation and Electrochemistry' and 'Inoculation Procedure' sections (SI Lines 178-185 and 192-202) to clarify this information as well. Because a pre-incubation step is not ideal for most applications, we are currently evaluating whether current decreases can be detected for naive cells induced in situ (e.g., not exposed to the inducing molecule beforehand).

How does timing/growth in the presence of the inducer affect the Figure 4a and b? Presumably protein expression is limited by the amount of inducer, but I do not see that data shown anywhere by measuring protein concentration or by measuring a different marker, like fluorescence. Was this published previously?

We have previously measured protein concentration, fluorescence (when expressing *sfgfp* or *eyfp*), and other important characterization metrics for the relevant genetic circuits in the following previous publications:

- Dundas, C. M., Walker, D. J., & Keitz, B. K. (2020). Tuning extracellular electron transfer by *Shewanella oneidensis* using transcriptional logic gates. *ACS synthetic biology*, 9(9), 2301-2315.
- Graham, A. J., Dundas, C. M., Hillsley, A., Kasprak, D. S., Rosales, A. M., & Keitz, B. K. (2020). Genetic control of radical cross-linking in a semisynthetic hydrogel. *ACS biomaterials science & engineering*, 6(3), 1375-1386.

A paper with more rigorous characterization of the 2-input gates is currently under peer-review, but has not been published yet.

- Graham, A. J., Dundas, C. M., Partipilo, G., Miniel Mahfoud, I. E., FitzSimons, T., Rinehart, R., ... & Keitz, B. K. (2021). Transcriptional Regulation of Synthetic Polymer Networks. *BioRxiv*, 2021-10.

We did not rigorously measure the effect of how long cells grow in the presence of an inducer on the rate of current decrease. However, as shown in Figure 4a and 4b, we did measure different inducer concentrations and have now included the raw curves for each of these points in the Supplementary Information (Figure S4).

We are currently investigating dynamic induction conditions where cells are only exposed to inducers at the time of inoculation into the OECT chamber and have had some initial success. We believe this is important as it allows for real-time, sensitive monitoring of EET gene expression dynamics. This method provides quantitative data on gene activation and is non-invasive, enabling continuous tracking without disrupting the cells. It's also an effective way to study the regulatory mechanisms of EET pathways. Currently, we tested a $\Delta mtrC$ strain with *mtrC* buffer gate under full induction strength (1 mM IPTG) alongside varying electron acceptor (fumarate) concentrations. For these experiments, cells were only exposed to IPTG upon inoculation into the device. As shown in the figure below, an increase in fumarate concentration correlates with a greater reduction in channel current I_{DS} . This trend suggests a concentration-dependent increase in residual EET activities in the mutants, in accordance with increasing cell growth and number with higher fumarate concentrations. Additionally, an initial plateau can be observed before the rapid I_{DS} decay. The initial plateaus were noted as X_0 in hours. For the induced $\Delta mtrC$ strain with buffer gate (Figure B), we observe an accelerated decay in I_{DS} , signifying enhanced reductive action upon induction. This is further quantified in Figure C, where the fitted rate constants are compared, demonstrating a clear relationship between IPTG induction and fumarate concentration with EET activities. The results show the feasibility of monitoring the dynamics of EET gene expression with OECTs in real time.

In future work, we intend to clarify the origins of the initial plateau observed in the current response. We will particularly assess if bacteria's preferential reduction of fumarate before interacting with the electrode material contributes to the delayed detection of EET gene expression. Specifically, we will identify optimal fumarate concentrations that minimize delay and examine alternative electron acceptors. Our goal is to refine the methodology for more direct and concurrent monitoring of the interactions between cells and OECTs and between cells and electron acceptors, thereby improving the real-time observation of EET gene expression.

REVIEWERS' COMMENTS

Reviewer #1 (Remarks to the Author):

The authors' responses to previous inquiries have remained inadequate. A notable deficiency lies in the elucidation of electron transfer mechanisms, with the authors asserting an electron transfer pathway involving direct contact with the electrode and the PEDOT:PSS channel. However, the use of specific strains that do not develop biofilm has been overlooked, raising pertinent doubts regarding the purported 'direct' contact. This underscores a potential misunderstanding by the authors not only of electron transfer mechanisms within a bacterial medium but also of the role played by ions and flavins. This inference is particularly derived from their utilization of an 'organic' electrochemical transistor system rather than conventional potentiometric/amperometric systems capable of distinguishing between ionic and redox contributions, a capability lacking in the suggested system.

Furthermore, the authors' disregard for the contribution of ions remains notable. Their statement referencing the use of abiotic controls, devoid of bacterial cells yet subjected to identical electrode bias voltages as the biotic samples and assessed alongside biotic OECTs, revealing typical OECT behavior (Figure 2f) attributed to ions from the growth medium traversing the PEDOT:PSS channel, fails to acknowledge the capacity of bacteria to facilitate ion exchange across their membranes, thereby modulating the overall ion concentration within the medium contingent upon prevailing conditions.

The contention that the Ag/AgCl electrode does not influence bacterial growth or act as a non-bactericidal agent, irrespective of continuous or intermittent usage, lacks substantial substantiation. Even brief exposure to such electrodes can act as a stressor on bacterial cultures, consequently impacting their growth dynamics. It is strongly recommended to consider alternative electrode materials. This argument partially extends to the PEDOT:PSS electrode, although the authors appear to acknowledge the advantages conferred by the application of such conductive polymers.

Based on the above points, the manuscript does not match the quality standards to be published in an important journal like Nature Communication, therefore it should be rejected directly.

Reviewer #2 (Remarks to the Author):

The authors have successfully addressed all my comments and the manuscript is now ready to be accepted.

Reviewer #3 (Remarks to the Author):

I do not have further comments. Well done

Reviewer #1 (Remarks to the Author):

The authors' responses to previous inquiries have remained inadequate. A notable deficiency lies in the elucidation of electron transfer mechanisms, with the authors asserting an electron transfer pathway involving direct contact with the electrode and the PEDOT:PSS channel. However, the use of specific strains that do not develop biofilm has been overlooked, raising pertinent doubts regarding the purported 'direct' contact. This underscores a potential misunderstanding by the authors not only of electron transfer mechanisms within a bacterial medium but also of the role played by ions and flavins.

Our experimental results indicate that direct contact electron transfer (EET) is the predominant mechanism in our OECT setup, supported by the data showing biofilm-deficient and flavin-deficient strains' behaviors. These observations suggest that while biofilms and flavins play a role in EET, their absence did not preclude electroactive bacteria from interacting with the OECT, as detailed in Figures 3b and 3c.

This inference is particularly derived from their utilization of an 'organic' electrochemical transistor system rather than conventional potentiometric/amperometric systems capable of distinguishing between ionic and redox contributions, a capability lacking in the suggested system.

One advantage of OECTs is that they are meant to couple ionic and redox transport processes. As mentioned in the manuscript, this gives them superior transconductance, sensing capabilities, and the potential to act as artificial synapses. There are ways to distinguish ionic and electronic contributions, but this was not the focus of our manuscript. We acknowledge the need for further molecular insights into the EET process across different conductive and semiconductive materials, which remains an open question in the field.

Furthermore, the authors' disregard for the contribution of ions remains notable. Their statement referencing the use of abiotic controls, devoid of bacterial cells yet subjected to identical electrode bias voltages as the biotic samples and assessed alongside biotic OECTs, revealing typical OECT behavior (Figure 2f) attributed to ions from the growth medium traversing the PEDOT:PSS channel, fails to acknowledge the capacity of bacteria to facilitate ion exchange across their membranes, thereby modulating the overall ion concentration within the medium contingent upon prevailing conditions.

Based on our initial response, we affirm that our experimental controls and subsequent observations underscore the minimal role of intracellular ion transport in OECT operation when compared to the significant contributions from metabolically active *S. oneidensis* cells. This conclusion is bolstered by our control experiments involving abiotic OECTs and supernatants from cell cultures, which did not exhibit notable current changes. Our analysis, including the comparison of EET-deficient strains with the wild-type strain (Figure 3d and 3e), further supports the premise that ionic changes at the cell membrane-channel interface are not the primary drivers of current modulation in OECTs. Finally, the far greater concentration of ions in bulk solution means that any ion transport into or out of cells has a negligible effect on bulk ion concentration.

The contention that the Ag/AgCl electrode does not influence bacterial growth or act as a non-bactericidal agent, irrespective of continuous or intermittent usage, lacks substantial substantiation. Even brief exposure to such electrodes can act as a stressor on bacterial cultures, consequently impacting their growth dynamics. It is strongly recommended to consider alternative electrode materials. This argument partially extends to the PEDOT:PSS electrode, although the authors appear to acknowledge the advantages conferred by the application of such conductive polymers.

We recognize concerns regarding the antimicrobial properties of Ag/AgCl electrodes. However, our experiments demonstrate that the transient presence of Ag/AgCl during measurements does not significantly influence the bacterial activity observed in our OECT setup (as current continues to decrease even after the transient inclusion of such electrodes). As mentioned in our previous response, Ag/AgCl electrodes were typically added at the very end of an experiment, after cell growth/metabolism were no longer relevant, in order to obtain absolute potential measurements. Finally, we rigorously examined cell viability and growth using PEDOT:PSS (Figure 1).

Based on the above points, the manuscript does not match the quality standards to be published in an important journal like Nature Communication, therefore it should be rejected directly.

We regret that our work did not meet the reviewer's expectations.